# A Simple Unified Uncertainty-Guided Framework for Offline-to-Online Reinforcement Learning

## Abstract

Offline reinforcement learning (RL) provides a promising solution to learning an agent fully relying on a data-driven paradigm. However, constrained by the limited quality of the offline dataset, its performance is often sub-optimal. Therefore, it is desired to further finetune the agent via extra online interactions before deployment. Unfortunately, offline-to-online RL can be challenging due to two main challenges: *constrained exploratory behavior* and *state-action distribution shift*. To this end, we propose a **S**imple **U**nified u**N**certainty-**G**uided (SUNG) framework, which naturally unifies the solution to both challenges with the tool of uncertainty. Specifically, SUNG quantifies uncertainty via a VAE-based state-action visitation density estimator. To facilitate efficient exploration, SUNG presents a practical optimistic exploration strategy to select informative actions with both high value and high uncertainty. Moreover, SUNG develops an adaptive exploitation method by applying conservative offline RL objectives to high-uncertainty samples and standard online RL objectives to low-uncertainty samples to smoothly bridge offline and online stages. SUNG achieves state-of-the-art online finetuning performance when combined with different offline RL methods, across various environments and datasets in D4RL benchmark.

## 1 Introduction

Offline reinforcement learning (RL) (Levine et al., 2020), which enables agents to learn from a fixed dataset without interacting with the environment, has demonstrated significant success in many tasks where interaction is expensive or risky (Burns et al., 2023; Lee et al., 2023; Zhao et al., 2023b). However, such a data-driven learning paradigm inherently limits the agent's performance as it fully relies on a typically sub-optimal dataset (Lee et al., 2022; Zhang et al., 2023). To overcome this issue, the RL community has been exploring the offline-to-online setting, which incorporates additional online interactions to further finetune the pretrained offline RL agents (Lee et al., 2022; Mao et al., 2022; Zhang et al., 2023; Zheng et al., 2022).

While the pretraining-finetuning paradigm in offline-to-online RL is natural and intuitive, delivering the expected online improvement can be challenging in practice due to two main challenges: (1) *Constrained exploratory behavior*: The conservative offline RL objectives, which require the agents to perform actions within the support of the dataset during pretraining, can constrain the agents' exploratory behavior for online interactions. As a result, they cannot achieve efficient exploration, and thus fail to fully benefit from the trial-and-error paradigm in online RL. (2) *State-action distribution shift*: During finetuning stage, agents may encounter unfamiliar state-action regimes that fall outside of the support of the dataset. Accordingly, the state-action distribution shift between offline and online data occurs, leading to the well-known extrapolation error (Fujimoto et al., 2019; Kumar et al., 2020) during exploitation, which may wipe out the good initialization obtained from the pretraining stage. Existing research typically focuses on tackling one aspect of the challenges above by developing either efficient exploration (Zheng et al., 2022; Mark et al., 2022) or effective exploitation (Lee et al., 2022; Zheng et al., 2023; Zhang et al., 2023), resulting in limited offline-to-online improvement. However, simultaneously and efficiently addressing both challenges brings additional difficulty: *aggressive exploration may exacerbate the distribution shift, while conservative exploitation can hinder agents from efficient online finetuning.*

In this work, we aim to provide a unified solution to both challenges for better sample efficiency during online finetuning. To achieve this, we recognize that both challenges are closely tied to appropriate identification and treatment of unseen state-action pairs. Specifically, efficient exploration necessitates discovery of informative unseen regions of state-action space. Furthermore, effective exploitation needs precise characterization of out-of-distribution (OOD) data to avoid either overly conservative or aggressive policy learning. As such, we leverage proper quantification and utilization of the uncertainty (Lee et al., 2021; Bai et al., 2022; An et al., 2021; Wu et al., 2021) to naturally address both challenges and achieve the desired trade-off between exploration and exploitation.

To this end, we present a **S**imple **U**nified u**N**certainty **G**uided (SUNG) framework, which provides a generic solution for offline-to-online RL to enable finetuning agents pretrained with different offline RL objectives. Unlike recent uncertainty-aware RL methods that rely on the ensemble technique for uncertainty quantification (Bai et al., 2022; An et al., 2021; Lee et al., 2021), SUNG utilizes a simple yet effective approach that estimates the state-action visitation density with a variational auto-encoder (VAE) (Kingma & Welling, 2013) to quantify the state-action uncertainty. In contrast to prior offline-to-online RL methods, SUNG simultaneously addresses both challenges by leveraging the tool of uncertainty. Concretely, we develop a practical optimistic exploration strategy that follows the principle of optimism in the face of uncertainty (Brafman & Tennenholtz, 2002; Audibert et al., 2009). The main idea here is to select those state-action pairs with both high value and high uncertainty for efficient exploration. We also propose an adaptive exploitation method to handle the state-action distribution shift by identifying and constraining OOD samples. The key insight is to leverage conservative offline RL objectives for high-uncertainty samples, and standard online RL objectives for low-uncertainty samples. This enables agents to smoothly adapt to changes in the state-action distribution. Notably, the insights of SUNG are generally applicable and can be combined with most model-free offline RL methods in principle. Empirically, SUNG benefits from consideration of uncertainty to guide both exploration and exploitation, thereby exceeding state-of-the-art methods on D4RL benchmarks (Fu et al., 2020), with 14.54% and 14.93% extra averaged offline-to-online improvement compared to the best baseline in MuJoCo and AntMaze domains, respectively. In addition, SUNG demonstrates remarkable robustness in finetuning performance across a range of hyper-parameters. Furthermore, we showcase SUNG's seamless integration with ensemble techniques, leading to superior finetuning performance.

**Summary of Contributions.** (1) We propose a generic framework SUNG for sample-efficient offline-to-online RL, which can be combined with existing offline RL methods. (2) We introduce an optimistic exploration strategy via bi-level action selection to select informative actions for efficient exploration. (3) We develop an adaptive exploitation method with OOD sample identification and regularization to smoothly bridge offline RL and online RL objectives. (4) Experimental results demonstrate that SUNG outperforms the previous state-of-the-art when combined with different offline RL methods, across various types of environments and datasets.

## 2 RELATED WORK

**Offline RL.** Offline RL (Levine et al., 2020) aims at learning a policy solely from a fixed offline dataset. Most prior works in offline RL have been dedicated to addressing the extrapolation error due to querying the value function with OOD actions (Fujimoto et al., 2019; Kumar et al., 2020). As such, it is critical to constrain the learned policy to perform actions within the support set of the offline dataset. Common strategies include policy constraint (Fujimoto et al., 2019; Kumar et al., 2019; Fujimoto & Gu, 2021; Wu et al., 2022), value regularization (Kumar et al., 2020; Lyu et al., 2022; Kostrikov et al., 2021; Bai et al., 2022), etc. However, previous works have shown that the the performance of offline RL agent is typically limited by the quality of the datasets (Nair et al., 2020; Lee et al., 2022), which prompts further investigation for the offline-to-online setting.

**Offline-to-Online RL.** Offline-to-online RL involves pretraining with offline RL and finetuning via online interactions. Some offline RL methods naturally support further online finetuning with continual offline RL objectives (Wu et al., 2022; Lyu et al., 2022; Kostrikov et al., 2022). However, they typically tend to be too conservative and yield limited performance improvement (Yu & Zhang, 2023). Besides, some offline-to-online RL approaches are designed for one specific offline RL method (Nakamoto et al., 2023; Luo et al., 2023; Guo et al., 2023; Ghosh et al., 2022; Hong et al., 2022; Swazinna et al., 2022). For example, ODT (Zheng et al., 2022) introduces the max-entropy

RL for finetuning DT (Chen et al., 2021). ABCR (Zhao et al., 2022) proposes to adaptively loosen the conservative objectives of TD3+BC (Fujimoto & Gu, 2021). In contrast, we focus on the generic offline-to-online RL framework that can be combined with different offline RL methods (Lee et al., 2022; Mark et al., 2022; Zheng et al., 2023; Zhang et al., 2023; Li et al., 2023; Zhao et al., 2023a). Prior works typically address the challenges of exploration limitation and state-action distribution shift. For the former, O3F (Mark et al., 2022) utilizes the knowledge in value functions to guide exploration, but it is not compatible with value regularization based offline RL methods. For the latter, BR (Lee et al., 2022) utilizes the ensemble technique together with a balanced replay buffer that prioritizes near-on-policy samples. APL (Zheng et al., 2023) proposes to take different advantages of offline data and online data for adaptive policy learning. PEX (Zhang et al., 2023) freezes the pretrained policies and trains a new policy from scratch using an adaptive exploration strategy to avoid erasing pre-trained policies. Unlike the aforementioned approaches, SUNG unifies the two challenges by emphasizing the proper estimation and utilization of uncertainty.

**Uncertainty for RL.** Uncertainty-aware RL has achieved notable success in both online and offline RL (Lockwood & Si, 2022). In terms of online RL, a prominent topic in exploration is the study of epistemic uncertainty, which leads to the development of the well-known principle of optimism in the face of uncertainty (Audibert et al., 2009; Ciosek et al., 2019; Chen et al., 2017; Lee et al., 2021). In the context of offline RL, both model-based (Kidambi et al., 2020; Guo et al., 2022; Yu et al., 2020; Nikulin et al., 2023; Tennenholtz & Mannor, 2022; Yang et al., 2022; Swazinna et al., 2021) and model-free (An et al., 2021; Bai et al., 2022; Wu et al., 2021; Ghasemipour et al., 2022) approaches can benefit from the consideration of uncertainty to identify and handle OOD actions. However, the aforementioned methods typically utilize the ensemble technique to estimate the uncertainty, which can be computationally expensive in terms of both time and training resources. In contrast, we adopt VAE (Kingma & Welling, 2013) for efficient and effective uncertainty quantification. Besides, we focus on the offline-to-online setting, which distinguishes SUNG from them.

## 3 Preliminaries

**RL.** We follow the standard RL setup that formulates the environment as a Markov decision process (MDP) $\mathcal{M} = (\mathcal{S}, \mathcal{A}, p, r, \gamma)$, where $\mathcal{S}$ is the state space, $\mathcal{A}$ is the action space, $p(s'|s, a)$ is the transition distribution, $r(s, a)$ is the reward function, and $\gamma \in [0, 1)$ is the discount factor. The goal of RL is to find a policy $\pi(a|s)$ that maximizes the expected return $\mathbb{E}_\pi[\sum_{t=0}^\infty \gamma^t r(s_t, a_t)]$.

**Off-policy RL.** Off-policy RL methods, such as TD3 (Fujimoto et al., 2018) and SAC (Haarnoja et al., 2018), have been widely applied due to their sample efficiency. These methods typically alternate between policy evaluation and policy improvement. In particular, given an experience replay dataset $\mathcal{D}$, TD3 learns a deterministic policy $\pi_\phi(s)$ and a state-action value function $Q_\theta(s, a)$, parameterized by $\phi$ and $\theta$, respectively. The value function can be updated via temporal difference (TD) learning as

$$\mathcal{L}_Q(\theta) = \mathbb{E}_{(s,a,r,s') \sim \mathcal{D}} \left[ (Q_\theta(s, a) - r - \gamma Q_{\bar{\theta}}(s', \pi_\phi(s')))^2 \right], \tag{1}$$

where $Q_{\bar{\theta}}$ is the target value network for stabilizing the learning process. Then, the policy can be updated to maximize the current Q value:

$$\mathcal{L}_\pi(\phi) = \mathbb{E}_{s \sim \mathcal{D}} \left[ -Q_\theta(s, \pi_\phi(s)) \right]. \tag{2}$$

**Offline RL.** In the offline RL setting, the agent only has access to a fixed dataset $\mathcal{D} = \{(s, a, r, s')\}$. Although off-policy RL methods can learn from data collected by any policy in principle, they fail in the offline RL setting. This can be attributed to the well-known extrapolation error (Fujimoto et al., 2019; Kumar et al., 2020) due to querying the value function with OOD actions. As such, one main line of model-free offline RL research is to constrain the learned policy to perform actions within the support of the dataset in different ways, such as policy constraint (Fujimoto & Gu, 2021; Wu et al., 2022), value regularization (Kumar et al., 2020; Kostrikov et al., 2021; Lyu et al., 2022) and etc. Among them, two representative offline RL methods are TD3+BC (Fujimoto & Gu, 2021) and CQL (Kumar et al., 2020). The former adds a behavior cloning (BC) regularization term to the standard policy improvement in TD3:

$$\mathcal{L}_\pi^{\text{TD3+BC}}(\phi) = \mathcal{L}_\pi(\phi) + \lambda_{\text{BC}} \mathbb{E}_{(s,a) \sim \mathcal{D}} \left[ (\pi_\phi(s) - a)^2 \right], \tag{3}$$

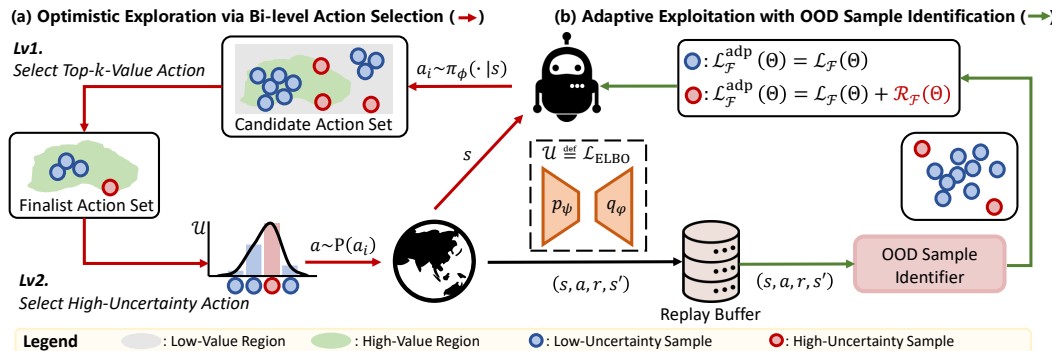

Figure 1: Overview of SUNG. During online finetuning, we alternate between **(a)** optimistic exploration strategy to collect behavior data from environment and **(b)** adaptive exploitation method to improve the policy. We adopt VAE for state-action density estimation to quantify uncertainty.

where $\lambda_{\mathrm{BC}}$ balances the standard policy improvement loss and BC regularization. In contrast, the latter resorts to pessimistic under-estimate of Q values during policy evaluation in SAC by

$$\mathcal{L}_Q^{\mathrm{CQL}}(\theta) = \mathcal{L}_Q(\theta) + \lambda_{\mathrm{CQL}} \left( \mathbb{E}_{s \sim \mathcal{D}, a \sim \pi_\phi} \left[ Q_\theta \left( s, a \right) \right] + \mathbb{E}_{(s,a) \sim \mathcal{D}} \left[ -Q_\theta \left( s, a \right) \right] \right), \quad (4)$$

where $\lambda_{\mathrm{CQL}}$ denotes the trade-off factor. Then, we can formally summarize both methods as

$$\mathcal{L}_{\mathcal{F}}^{\mathrm{offline}}(\Theta) = \mathcal{L}_{\mathcal{F}}(\Theta) + \lambda \mathcal{R}_{\mathcal{F}}(\Theta), \quad (5)$$

where $\mathcal{F}$ represents either a policy or a value function, parameterized by $\Theta$, $\mathcal{R}$ denotes a regularizer to prevent the extrapolation error, and $\lambda$ balances standard loss and regularization. Note that most model-free offline RL methods can be summarized as Eq. (5) in an explicit (Kumar et al., 2020; Fujimoto & Gu, 2021; Kostrikov et al., 2021; Wu et al., 2022; Lyu et al., 2022) or implicit manner (Kumar et al., 2019; Rezaeifar et al., 2022; Kostrikov et al., 2022; Bai et al., 2022; Xu et al., 2023), which motivates the adaptive exploitation in Section 4.3.

# 4 SUNG

In this section, we present a simple unified uncertainty-guided framework (SUNG) for offline-to-online RL, as depicted in Fig. 1. Concretely, SUNG quantifies the state-action uncertainty by estimating a density function with VAE. Then, under the guidance of uncertainty, SUNG incorporates an optimistic exploration strategy and an adaptive exploitation method for online finetuning. Putting everything together, we summarize the full framework in Algorithm 1 and provide two implementation cases for bridging TD3+BC and TD3, and bridging CQL and SAC in Appendix A.

## 4.1 UNCERTAINTY QUANTIFICATION WITH DENSITY ESTIMATION

Many prior works typically utilize Q ensembles to qualify uncertainty (Wu et al., 2021; Mark et al., 2022; Ciosek et al., 2019; Lee et al., 2021). However, the ensemble technique may significantly increase computational costs. Thus, in this work, we utilize a simple yet effective approach by adopting VAE (Kingma & Welling, 2013) as a state-action visitation density estimator for uncertainty quantification. Specifically, given an encoder $q_\varphi(z|s,a)$ and a decoder $p_\psi(s,a|z)$ with parameters $\varphi$ and $\psi$, respectively, we can optimize them with the evidence lower bound (ELBO)

$$\mathcal{L}_{\mathrm{ELBO}}(s, a; \psi, \varphi) = \mathbb{E}_{q_\varphi(z|s,a)} \left[ -\log p_\psi(s, a|z) \right] + D_{\mathrm{KL}} \left[ q_\varphi(z|s,a) || p(z) \right], \quad (6)$$

where $p(z) = \mathcal{N}(0, I)$ is a fixed prior. Then, we utilize the negative log likelihood of the state-action density for uncertainty quantification, i.e., $\mathcal{U}(s, a) \overset{\mathrm{def}}{=} -\log p(s, a) \approx \mathcal{L}_{\mathrm{ELBO}}(s, a; \psi, \varphi)$, as we know $\log p(s, a) = -\mathcal{L}_{\mathrm{ELBO}}(s, a; \psi, \varphi) + D_{\mathrm{KL}}[q_\varphi(z|s,a) || p_\psi(z|s,a)]$. Intuitively, both the reconstruction loss and KL divergence in ELBO indicate the uncertainty for state-action space, i.e., whether a state-action pair can be well captured by the learned distribution.

Although it can be theoretically shown that a bias exists between ELBO and the negative log likelihood of the state-action density, a prior work has shown that such approximation is empirically

valid (Wu et al., 2022). Thus, we utilize ELBO for uncertainty quantification in practice, and leave the exploration of importance sampling techniques (Kingma et al., 2019; Rezende et al., 2014) for further bias reduction as future work.

## 4.2 OPTIMISTIC EXPLORATION VIA BI-LEVEL ACTION SELECTION

A main line of model-free offline RL methods perform conservative objectives to penalize OOD actions during offline pretraining. However, such conservative objectives inherently limit agents' exploration power, which impedes agents to fully benefit from online finetuning (Mark et al., 2022). To overcome this limitation, our goal is to properly measure the informativeness of unseen actions, and then select the most informative action for exploration. To achieve this, we propose an optimistic exploration strategy based on the principle of optimism in the face of uncertainty (Brafman & Tennenholtz, 2002; Audibert et al., 2009). The key insight is that an informative action is expected to possess both high Q value and high uncertainty.

Existing works typically estimate epistemic uncertainty via Q ensembles and derive an upper confidence bound (UCB) to direct the exploration (Chen et al., 2017; Ciosek et al., 2019; Lee et al., 2021). Different from them, we extend this idea by utilizing $\mathcal{U}(s, a)$ for uncertainty quantification. To avoid inaccurate estimates for Q value and uncertainty, we only select near-on-policy actions for exploration. Consequently, the objective of the exploration policy $\pi_E(s)$ is to maximize the informativeness while remaining near-on-policy, i.e.,

$$\pi_E(s) = \arg\max_{a \in \mathcal{A}} Q_\theta(s, a) + \beta \mathcal{U}(s, a),$$
$$\text{s.t. } ||a - \pi_\phi(s)||^2 \leq \delta, \tag{7}$$

where $\beta$ controls the optimism level and $\delta$ controls the on-policyness, given the assumption of smoothness in the physical transition model. As such, the derived behavior policy can not only increase the chance of executing informative actions according to the principal of optimism in the face of uncertainty, but also guarantee the on-policyness to ensure the stability.

However, such an optimization problem is impractical to solve due to two main challenges. On the one hand, it is not straightforward to find optimal action in high-dimensional continuous action space, which is impossible to enumerate exhaustively. On the other hand, it is hard to set a proper value for optimism level $\beta$ because Q value and uncertainty have different value ranges across different tasks, which necessitates task-specific hyper-parameter tuning.

To mitigate both challenges above, we propose a simple yet effective bi-level action selection mechanism. First, we generate a candidate action set $\mathcal{C} = \{a_i\}_{i=1}^N$ with $N$ candidate actions. Here, the action is generated by sampling $a_i \sim \pi_\phi(\cdot|s)$ for stochastic policy, while by adding sampled Gaussian noise $a_i = \pi_\phi(s) + \epsilon_i, \epsilon_i \sim \mathcal{N}(0, \delta)$ for deterministic policy. Then, we rank the candidate actions according to the Q-values (or uncertainty) to select top-$k$ candidate actions as a finalist action set $\mathcal{C}_f$. Finally, we construct a categorical distribution according to the uncertainty (or Q values) to select the action from $\mathcal{C}_f$ for interacting with the environment:

$$\mathrm{P}(a_i) := \frac{\exp\left(\mathcal{U}(s, a_i)/\alpha\right)}{\sum_j \exp\left(\mathcal{U}(s, a_j)/\alpha\right)}, \forall i \in [1, ..., k], \tag{8}$$

where $\alpha$ is the softmax temperature. By altering ranking criteria for the finalist action set and $k$, we can flexibly adjust the preference for high Q value or high uncertainty to achieve the desired trade-off. As discussed in Appendix B, value regularization based methods may diverge when preference for Q value is present. To address this, we establish the ranking criteria for the finalist action set as uncertainty for value regularization-based methods and as Q value for other offline RL methods.

The optimistic exploration strategy enables agents to interact the environment with informative actions, thereby boosting the sample efficiency of offline-to-online RL. However, it may also bring negative effects by further increasing state-action distribution shift, as a result of the principle of optimism in the face of uncertainty. To this end, we introduce an adaptive exploitation method in the following subsection, which aims to mitigate the state-action distribution shift.

### 4.3 ADAPTIVE EXPLOITATION WITH OOD SAMPLE IDENTIFICATION

Action distribution shift poses a significant challenge for offline RL algorithms (Fujimoto et al., 2019; Kumar et al., 2019; Levine et al., 2020; Kumar et al., 2020), as the bootstrapping term in policy evaluation involves actions derived from the learned policy $\pi_\phi$. This can result in the extrapolation error due to querying Q functions with OOD actions, leading to biased policy improvement towards these OOD actions with erroneously high Q values. Note that model-free offline RL methods do not suffer from state distribution shift during training, since policy evaluation only queries Q functions with states present in the offline dataset.

However, during online finetuning, both state and action distribution shifts occur since agents may encounter unfamiliar state-action regimes that fall outside of the support of the dataset. Worse still, as the proposed optimistic exploration strategy follows the principle of optimism in the face of uncertainty, the state-action distribution shift may be further exacerbated. Preliminary experiments from (Lee et al., 2022; Mark et al., 2022; Zheng et al., 2023) show that this may erase the good initialization obtained from offline pretraining.

To tackle the state-action distribution shift issue, we propose an adaptive exploitation method with OOD sample identification. The underlying idea is developed from the observations in previous works (Lee et al., 2022; Mark et al., 2022; Zheng et al., 2023) that finetuning with continual offline RL objectives typically derives stable but limited performance, while finetuning with online RL objectives typically derives unstable performance due to the distribution shift caused by OOD state-action pairs. Therefore, we propose a novel approach that leverages the benefits of both objectives by utilizing conservative offline RL objectives for state-action pairs with high uncertainty while aggressive online RL objectives for state-action pairs with low uncertainty. As shown in Eq. (5), offline RL objectives consist of a standard online RL objective $\mathcal{L}_\mathcal{F}(\Theta)$ and a regularizer $\mathcal{R}_\mathcal{F}(\Theta)$. Thus, we can derive the objectives for adaptive exploitation by introducing an uncertainty-guided OOD sample identifier $\mathcal{I}(s, a)$:

$$\mathcal{L}_\mathcal{F}^{\mathrm{adp}}(\Theta) = \mathcal{L}_\mathcal{F}(\Theta) + \lambda \mathcal{I}\left(s, \pi_\phi\left(s\right)\right) \mathcal{R}_\mathcal{F}(\Theta). \tag{9}$$

We expect that $\mathcal{I}(s, a)$ is a large value for state-action pairs with high uncertainty and a small value for those with low uncertainty. Therefore, we construct such an identifier using uncertainty estimation $\mathcal{U}(s, a)$ as below. For a minibatch of state-action pairs $\{s_i, a_i\}_{i=1}^M$, we select the top $p\%$ of them as OOD state-action pair sets $\mathcal{D}_{\mathrm{OOD}}$, according to its corresponding uncertainty estimation $\mathcal{U}(s_i, a_i)$. The selection process can be performed by sampling from a categorical distribution similar to Eq. (8). Then, out of simplicity, we define the OOD sample identifier as

$$\mathcal{I}(s, a) := \left\{ \begin{array}{ll} 1, & \mathrm{if} (s, a) \in \mathcal{D}_{\mathrm{OOD}}, \\ 0, & \mathrm{else}. \end{array} \right. \tag{10}$$

By setting $p = 0$, the training objective is precisely the online RL objective, while setting $p = 100$ results in the offline RL objective. Thus, by tuning the value of $p$, the proposed adaptive exploitation method can attain the desired trade-off between performance and stability.

## 5 EXPERIMENTS

### 5.1 EXPERIMENTAL SETUP

**Settings.** We evaluate on the D4RL benchmark (Fu et al., 2020), which provides various continuous-control tasks and datasets. We focus on MuJoCo and AntMaze domains. We first perform 1M gradient steps for offline pretraining, and then perform 100K environment steps for online finetuning. While some prior works (Kostrikov et al., 2022; Zhang et al., 2023; Zheng et al., 2023) take 1M environment steps for finetuning, we argue that 1M environment steps are even enough for an online RL agent to achieve expert-level performance. Thus, we believe finetuning for 100K environment steps is a more reasonable setting, which is also adopted by (Lyu et al., 2022; Zheng et al., 2023).

**Backbone Offline RL Methods.** To evaluate the generality of SUNG, we choose two representative offline RL methods as the backbone for pretraining, i.e., TD3+BC (Fujimoto & Gu, 2021) and CQL (Kumar et al., 2020). Concretely, TD3+BC extends TD3 with behavior cloning based policy constraint, while CQL extends SAC with value regularization. For AntMaze domains, we substitute

Table 1: Comparison of the averaged D4RL score on MuJoCo tasks with **TD3+BC** as the offline RL backbone method. We report the mean and standard deviation over 5 seeds.

| | TD3+BC | offline-ft | online-ft | BR | O3F | APL | PEX | PROTO | SUNG |
|---|---|---|---|---|---|---|---|---|---|
| **halfcheetah-r-v2** | 11.5 | 34.3±2.4 | 57.7±1.6 | 67.6±11.9 | 71.4±3.3 | 70.0±4.7 | 53.4±9.1 | 50.7±1.3 | **76.6±2.0** |
| **hopper-r-v2** | 8.7 | 8.2±0.2 | 11.2±1.9 | 25.7±8.7 | 11.7±2.0 | 27.1±14.0 | 37.4±10.6 | 13.4±9.5 | **38.7±15.0** |
| **walker2d-r-v2** | 5.4 | 7.0±4.1 | 6.2±3.9 | 9.9±4.6 | 11.6±5.1 | 13.8±4.0 | **33.1±16.4** | 3.6±3.1 | 14.1±5.1 |
| **halfcheetah-m-v2** | 48.0 | 49.3±0.4 | 67.6±2.4 | 79.5±7.4 | 77.8±1.1 | **80.9±2.0** | 52.3±21.1 | 67.4±1.8 | 80.7±2.5 |
| **hopper-m-v2** | 61.5 | 58.8±3.9 | 78.3±32.7 | 93.1±10.8 | **102.0±2.0** | 76.9±24.2 | 73.9±17.8 | 60.5±23.3 | 101.8±6.0 |
| **walker2d-m-v2** | 82.2 | 84.6±1.2 | 68.2±11.5 | 70.1±21.4 | 97.1±2.2 | 98.2±13.5 | 56.7±28.5 | 79.5±9.4 | **113.5±1.9** |
| **halfcheetah-m-r-v2** | 44.6 | 47.0±0.9 | 66.3±1.0 | 65.0±14.6 | 67.6±2.9 | **71.5±1.3** | 53.2±9.8 | 61.0±1.7 | 69.7±3.4 |
| **hopper-m-r-v2** | 55.9 | 85.4±7.6 | 89.9±13.5 | 97.2±13.9 | 97.6±4.9 | 100.6±9.8 | 90.8±18.7 | 100.4±1.0 | **101.3±7.0** |
| **walker2d-m-r-v2** | 71.7 | 80.2±8.7 | 87.4±4.0 | 82.7±17.8 | 100.9±3.7 | 108.2±3.6 | 70.6±13.3 | 93.7±3.3 | **109.2±1.9** |
| **Total** | 389.6 | 454.9 | 532.9 | 590.9 | 637.7 | 647.1 | 521.2 | 530.2 | **705.7** |

Table 2: Comparison of the averaged D4RL score on MuJoCo tasks with **CQL** as the offline RL backbone method. We report the mean and standard deviation over 5 seeds. Note that O3F is omitted due to its divergence in this setting. Refer to Appendix B for detailed explanations.

| | CQL | offline-ft | online-ft | BR | APL | PEX | PROTO | SUNG |
|---|---|---|---|---|---|---|---|---|
| **halfcheetah-r-v2** | 23.5 | 28.8±2.3 | 50.2±1.5 | 61.8±12.3 | 67.7±9.6 | 50.6±2.2 | 24.9±7.9 | **69.1±9.2** |
| **hopper-r-v2** | 6.4 | 31.2±0.5 | 28.3±6.2 | 23.8±7.9 | 41.8±22.0 | 34.3±8.9 | 30.1±3.1 | **44.3±11.7** |
| **walker2d-r-v2** | 4.5 | 5.6±3.4 | 8.2±5.6 | 4.0±2.3 | 6.3±1.8 | 10.7±2.8 | 1.6±2.3 | **14.5±6.1** |
| **halfcheetah-m-v2** | 48.1 | 48.9±0.2 | 52.1±25.6 | 56.7±28.5 | 44.7±38.5 | 43.5±2.4 | 52.1±26.7 | **79.7±1.0** |
| **hopper-m-v2** | 73.7 | 74.1±1.4 | 91.0±10.4 | 97.7±3.7 | 102.7±3.1 | 46.3±15.1 | 98.8±5.1 | **104.1±1.3** |
| **walker2d-m-v2** | 84.3 | 83.5±0.7 | 85.6±7.6 | 81.7±14.0 | 75.3±25.7 | 34.0±17.3 | 78.9±11.2 | **86.0±12.6** |
| **halfcheetah-m-r-v2** | 46.9 | 49.5±0.3 | 61.3±1.0 | 64.9±5.8 | **78.6±1.2** | 45.5±1.7 | 62.1±4.2 | 75.6±1.9 |
| **hopper-m-r-v2** | 96.0 | 95.0±1.0 | 92.8±19.5 | 88.5±21.8 | 97.4±9.5 | 66.5±24.2 | 92.6±19.4 | **101.9±9.1** |
| **walker2d-m-r-v2** | 83.4 | 84.5±1.0 | 86.9±12.4 | 78.8±28.0 | 103.2±19.0 | 40.1±17.9 | 94.8±1.8 | **108.2±4.2** |
| **Total** | 466.9 | 501.1 | 556.4 | 558.0 | 617.8 | 371.5 | 536.1 | **683.4** |

TD3+BC with SPOT (Wu et al., 2022) due to its inferior performance. SPOT is another extension of TD3 but with density guided policy constraint.

**Baselines.** We compare SUNG with the following baselines: **(1) offline-ft** leverages online interactions by performing offline RL objectives. **(2) online-ft** leverages online interactions by performing online RL objectives that correspond to the ones used for pretraining. **(3) BR** (Lee et al., 2022) prioritizes near-on-policy transitions from the replay buffer. **(4) O3F** (Mark et al., 2022) utilizes knowledge contained in value functions to guide exploration. **(5) APL** (Zheng et al., 2023) performs adaptive policy learning by incorporating different advantages of offline and online data. **(6) PEX** (Zhang et al., 2023) trains a new policy from scratch by utilizing the pre-trained policy for exploration. **(7) PROTO** (Li et al., 2023) introduces an iterative policy regularization scheme to achieve stable finetuning performance. Note that we also include IQL (Kostrikov et al., 2022), ODT (Zheng et al., 2022) and ACA (Yu & Zhang, 2023) for comparison with a seperate subsection in Appendix D.2. See Appendix C for further implementation details.

## 5.2 COMPARISONS ON D4RL BENCHMARKS

**Results on MuJoCo.** Results for MuJoCo domains with TD3+BC and CQL as backbone offline RL method are shown in Table 1 and Table 2, respectively. We remark that SUNG substantially outperforms previous state-of-the-art methods, exhibiting an additional 15.04% and 14.05% offline-to-online improvement over the best-performing baseline when combined with TD3+BC and CQL, respectively. This demonstrates the necessity of proper estimation and utilization of uncertainty for tackling both constrained exploratory behavior and state-action distribution shift. Moreover, this also verifies the generality of SUNG in facilitating online improvement for agents pretrained with different offline RL methods.

**Results on AntMaze.** We also provide results for AntMaze domains with SPOT and CQL as backbone offline RL method. We defer detailed results and analyses to Appendix D.1. We highlight that SUNG exhibits an additional 12.43% and 17.43% offline-to-online improvement over the best-performing baseline when combined with SPOT and CQL, respectively.

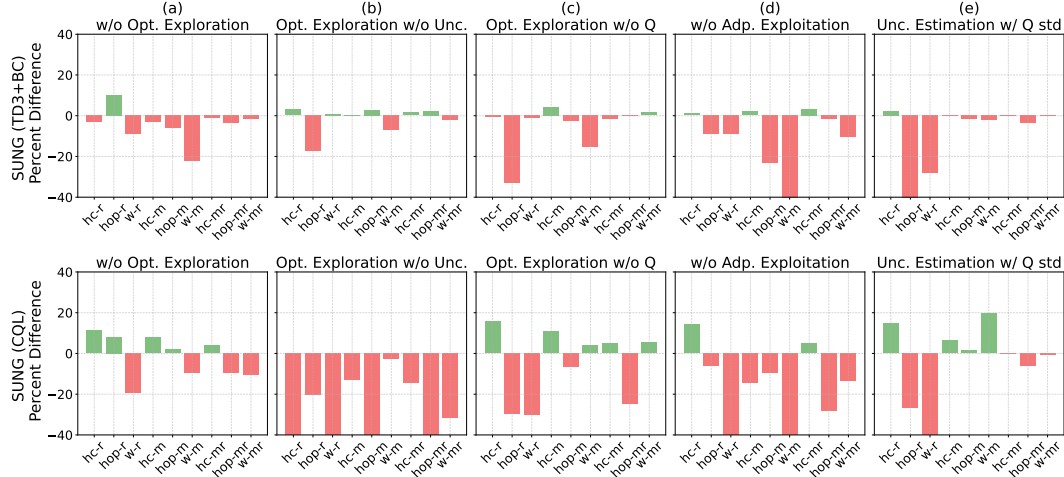

Figure 2: Performance difference of an ablation study of SUNG combined with TD3+BC and CQL, compared with the full algorithm. Opt. = Optimistic, Unc. = Uncertainty, Adp. = Adaptive. hc = HalfCheetah, hop = Hopper, w = Walker, r = random, m = medium, mr = medium-replay.

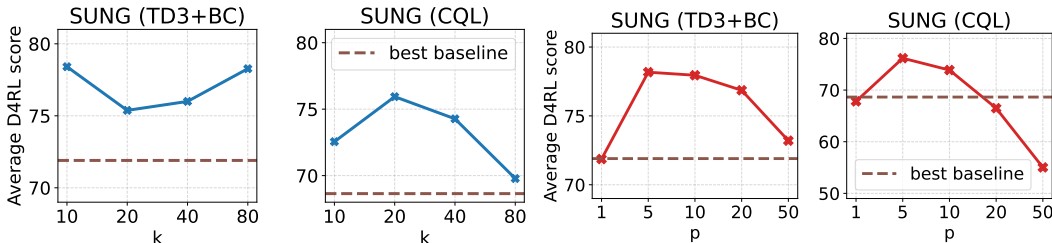

Figure 3: Comparing performance with different finalist action set size $k$ used in optimistic exploration.

Figure 4: Comparing performance with varying percent $p$ of identified OOD samples from the mini-batch used in adaptive exploitation.

## 5.3 ABLATION STUDIES

In this subsection, we perform an ablation study over the components in SUNG in Fig. 2. Refer to Appendix D.5 for implementation details and extra ablation studies.

**Ablation on Optimistic Exploration.** First, we evaluate **(a)** SUNG without optimistic exploration strategy, and find that the removal brings significant negative effects when combined with TD3+BC, but no remarkable effects when combined with CQL. The reason for this discrepancy is that CQL is built on top of SAC, which can gradually recover exploration power through max-entropy RL framework during online finetuning. Then, we ablate **(b)** uncertainty and **(c)** Q value in the optimistic exploration strategy to derive two variant strategies - one greedy for Q value and another greedy for uncertainty. As expected, both ablations degrade the performance to varying degrees, underscoring the significance of both Q value and uncertainty for exploration. Notably, the removal of uncertainty in SUNG when combined with CQL causes significant performance degradation, which can be attributed to similar reason detailed in Appendix B.

**Ablation on Adaptive Exploitation.** We evaluate **(d)** SUNG without the adaptive exploitation method, and observe that performance for finetuning TD3+BC and CQL significantly deteriorates on most tasks. This emphasizes the necessity of addressing the state-action distribution shift issue.

**Ablation on Uncertainty Quantification.** Finally, we **(e)** replace the VAE-based density estimator with the standard deviation of double Q values for uncertainty quantification (Ciosek et al., 2019). As expected, the results show that such replacement leads to performance deterioration on most tasks, since it is not sufficient to provide reliable uncertainty quantification. While utilizing the

ensemble technique can eliminate this issue (Lee et al., 2021; Chen et al., 2017), it can significantly increase computational costs. We leave the exploration of alternative methods as future work.

### 5.4 HYPER-PARAMETER ANALYSIS

**Analysis of Optimistic Exploration.** First, we investigate the finalist action set size $k$ in optimistic exploration. The aggregated results averaged over MuJoCo domains are shown in Fig. 3. As expected in Appendix B, SUNG with TD3+BC can accommodate preference for both high uncertainty and high value, whereas SUNG with CQL can only accommodate preference for high uncertainty. We remark that SUNG consistently outperforms the best baseline with any choice of $k$, which further verifies the superiority of SUNG.

**Analysis of Adaptive Exploitation.** Moreover, we explore the choice of percent $p$ of identified OOD samples from the mini-batch in adaptive exploitation. The aggregated results averaged over MuJoCo domains are shown in Fig. 4. Predictably, setting $p$ too conservatively or too aggressively may adversely degrade the performance. We find that $p = 5$ performs the best when combined with either TD3+BC or CQL, achieving the desired trade-off between performance and stability.

### 5.5 SUNG WITH ENSEMBLE TECHNIQUE

Previous works have shown that ensemble based offline RL methods can achieve robust pretraining and finetuning performance. In this subsection, we aim to illustrate that SUNG can be seamlessly compatible with the ensemble technique, leading to improved finetuning performance. Specifically, we follow previous works (Lee et al., 2022; Ball et al., 2023) to utilize CQL-10, a variant of CQL employing 10 Q functions, for offline pretraining. We compare SUNG against two baselines: **(1) BRPQ** (Lee et al., 2022) introduces both prioritized replay buffer and pessimistic Q ensembles for offline-to-online RL. **(2) RLPD** (Ball et al., 2023) is a recent state-of-the-art method that utilizes offline data

Table 3: Comparison of the averaged D4RL score on MuJoCo tasks with **CQL-10** as the offline RL backbone method. We report the mean and standard deviation over 5 seeds.

| | CQL-10 | BRPQ | RLPD (UTD=20) | SUNG (UTD=1) | SUNG (UTD=5) |
|---|---|---|---|---|---|
| **halfcheetah-r-v2** | 29.9 | 85.8±17.2 | 72.5±5.9 | **92.7**±4.2 | **97.5**±5.0 |
| **hopper-r-v2** | 7.4 | 28.4±12.0 | 87.8±14.0 | 62.5±13.8 | **89.7**±28.4 |
| **walker2d-r-v2** | 21.6 | 16.1±4.0 | 65.7±16.4 | 33.0±19.6 | **73.1**±33.0 |
| **halfcheetah-m-v2** | 55.0 | 80.3±20.5 | 84.2±2.2 | **96.6**±1.0 | **105.5**±3.6 |
| **hopper-m-v2** | 66.9 | 79.5±19.0 | 98.1±12.2 | **111.4**±0.8 | 86.3±24.2 |
| **walker2d-m-v2** | 83.2 | 74.4±13.9 | **114.3**±2.2 | 113.8±2.6 | 111.3±25.3 |
| **halfcheetah-m-r-v2** | 52.6 | 74.8±20.7 | 80.3±1.9 | **92.5**±0.5 | **96.7**±6.1 |
| **hopper-m-r-v2** | 102.3 | 71.6±10.1 | 75.0±17.9 | **100.7**±13.6 | 64.9±14.8 |
| **walker2d-m-r-v2** | 82.1 | 71.7±11.2 | 108.5±2.8 | 105.1±14.6 | **117.3**±5.8 |
| **Total** | 501.0 | 582.6 | 786.4 | **808.2** | **842.3** |

to accelerate online RL, benefiting from both the ensemble technique and high update-to-data (UTD) ratio. For a fair comparison, we present results of SUNG with UTD ratio of 1 and 5. Furthermore, in the Appendix D.3, we provide an additional comparison with the most recent ensemble-based offline-to-online RL method E2O (Zhao et al., 2023a).

As demonstrated in Table 3, SUNG consistently outperforms BRPQ and achieves results that are either superior or at least comparable to those of RLPD, which utilizes a high UTD ratio. Furthermore, it is worth noting that SUNG with UTD ratio of 1 exhibits weak results for hopper-r and walker2d-r, suggesting that poorly initialized policies are challenging to recover through low UTD ratio online finetuning. In contrast, SUNG with UTD ratio of 5 demonstrates competitive results in these settings. However, we notice that SUNG with UTD ratio of 5 shows subpar performance in hopper-m and hopper-m-r, possibly due to the catastrophic overestimation. One potential remedy, as suggested in RLPD (Ball et al., 2023), is layer normalization (Ba et al., 2016). We consider this avenue for future work.

## 6 CONCLUSION

This paper studies how to efficiently finetune pretrained offline RL agents via online interactions. We present SUNG, a simple unified uncertainty-guided framework, to tackle both challenges of constrained exploratory behavior and state-action distribution shift. SUNG leverages the quantified uncertainty to guide optimistic exploration and adaptive exploitation. Empirical results across different backbone offline RL methods, environments and datasets verify the superiority of SUNG.

## REPRODUCIBILITY STATEMENT

**(This section does not count towards the page limit.)**

We provide the detailed algorithm description in Appendix A and experimental implementation details in Appendix C. We will make our codes and pretrained checkpoints publicly available to facilitate the replication and verification of our results upon publication.

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

APPENDIX

## A IMPLEMENTATION DETAILS OF SUNG

In this section, we first present full algorithm for SUNG, and then provide two implementation cases of SUNG: one bridges TD3+BC (Fujimoto & Gu, 2021) and TD3 (Fujimoto et al., 2018), and another bridges CQL (Kumar et al., 2020) and SAC (Haarnoja et al., 2018). Note that existing offline RL methods are typically built on top of online off-policy RL methods. Thus, we utilize the corresponding online RL objectives for online finetuning to keep consistent.

### A.1 ALGORITHM IMPLEMENTATION DETAILS

**Offline-to-Online Replay Buffer.** Previous studies (Lee et al., 2022; Zheng et al., 2023) have highlighted the significance of incorporating both offline and online data during finetuning. Here, we adopt the offline-to-online replay buffer proposed in (Zheng et al., 2023) due to its simplicity. Specifically, OORB consists of two replay buffers: a fully online buffer that collects transitions from online interactions, and an offline buffer that stores transitions from both offline dataset and online interactions. During training, we randomly sample transitions from OORB with a Bernoulli distribution, where the probability of sampling from the fully online buffer is $p_{\text{OORB}}$, and the probability of sampling from the offline buffer is $1 - p_{\text{OORB}}$.

**SUNG Framework for Offline-to-Online RL.** Putting everything together, we summarize the full framework in Algorithm 1. SUNG first pretrains agents and VAE using offline dataset. Then, it turns to online finetuning with the proposed optimistic exploration strategy and adaptive exploitation method.

---

**Algorithm 1** SUNG: A Simple Unified Uncertainty-Guided Framework for Offline-to-Online RL

---

1: **Input:** Dataset $\mathcal{D}$, Offline RL algorithm $\{\mathcal{L}_Q^{\text{offline}}(\theta), \mathcal{L}_\pi^{\text{offline}}(\phi)\}$
2: Initialize Q network $Q_\theta$, policy network $\pi_\phi$, VAE network $p_\psi$ and $q_\varphi$
3: **while** in *offline pretraining phase* **do**
4:     Sample minibatch of transitions $(s, a, r, s') \sim \mathcal{D}$
5:     Update $\theta$ and $\phi$ minimizing $\mathcal{L}_Q^{\text{offline}}(\theta)$ and $\mathcal{L}_\pi^{\text{offline}}(\phi)$
6:     Update $\psi, \varphi$ minimizing $\mathcal{L}_{\text{ELBO}}(s, a; \psi, \varphi)$ in Eq. (6)
7: **end while**
8: Initialize OORB with $\mathcal{B}_{\text{OORB}} \leftarrow \{\mathcal{B} = \emptyset, \mathcal{D}\}$
9: **while** in *online finetuning phase* **do**
10:     // **Optimistic Exploration**
11:     Generate candidate action set $\mathcal{C} = \{a_i\}_{i=1}^N$
12:     Select actions with top-$k$ Q-value (or uncertainty) as finalist action set $\mathcal{C}_f$
13:     Sample the action $a_t \sim \text{P}(a_i), a_i \in \mathcal{C}_f$ in Eq. (8) according to the uncertainty (or Q value)
14:     Execute the action $a_t$, collect $s_{t+1}$ and $r_t$ from the environment
15:     Store transitions $(s_t, a_t, r_t, s_{t+1})$ into $\mathcal{B}$ and $\mathcal{D}$
16:     // **Adaptive Exploitation**
17:     Sample minibatch of transitions $(s, a, r, s') \sim \mathcal{B}_{\text{OORB}}$
18:     Update $\theta$ and $\phi$ minimizing $\mathcal{L}_{\mathcal{F}}^{\text{adp}}(\Theta)$ in Eq. (9)
19:     Update $\psi, \varphi$ minimizing $\mathcal{L}_{\text{ELBO}}(s, a; \psi, \varphi)$ in Eq. (6)
20: **end while**

---

### A.2 SUNG (TD3+BC): BRIDGING TD3+BC AND TD3

First, we show how SUNG (TD3+BC) bridges TD3+BC and TD3 to finetune the deterministic policy $\pi_\phi(s)$ and two Q value functions $Q_{\theta_1}(s, a), Q_{\theta_2}(s, a)$.

In terms of exploration, the behavior policy of SUNG (TD3+BC) is compatible with preference for either high Q value or high uncertainty, as demonstrated in the main body. Here, we take preference

---

**Algorithm 2** SUNG (TD3+BC): Bridging TD3+BC and TD3

---

1: **Input:** Dataset $\mathcal{D} = \{(s, a, r, s')\}$
2: **// Offline Pretraining**
3: Initialize Q network $Q_\theta$, policy network $\pi_\phi$, VAE network with $p_\psi$ and $q_\varphi$
4: **for** $t = 1$ **to** $T_1$ **do**
5:     Sample minibatch of transitions $(s, a) \sim \mathcal{D}$
6:     Update $\theta$ and $\phi$ minimizing $\mathcal{L}_Q(\theta)$ in Eq. (1) and $\mathcal{L}_\pi^{\text{TD3+BC}}(\phi)$ in Eq. (3)
7:     Update $\psi, \varphi$ minimizing $\mathcal{L}_{\text{ELBO}}(s, a; \psi, \varphi)$ in Eq. (6)
8: **end for**
9: **// Online Finetuning**
10: Initialize OORB with $\mathcal{B}_{\text{OORB}} \leftarrow \{\mathcal{B} = \emptyset, \mathcal{D}\}$
11: **for** $t = 1$ **to** $T_2$ **do**
12:     **// Optimistic Exploration**
13:     Generate candidate action set $\mathcal{C} = \{a_i\}_{i=1}^N$, where $a_i = \pi_\phi(s_t) + \epsilon_i, \epsilon_i \sim \mathcal{N}(0, \delta)$
14:     Select actions with top-$k$ Q-values as finalist action set $\mathcal{C}_f$
15:     Sample the action $a_t \sim \mathrm{P}(a_i), a_i \in \mathcal{C}_f$ in Eq. (8) according to the uncertainty
16:     Execute action $a_t$, collect $s_{t+1}$ and $r_t$ from the environment
17:     Store transitions $(s_t, a_t, r_t, s_{t+1})$ into $\mathcal{B}$ and $\mathcal{D}$
18:     **// Adaptive Exploitation**
19:     Sample minibatch of transitions $(s, a, r, s') \sim \mathcal{B}_{\text{OORB}}$
20:     Construct the OOD identifier $\mathcal{I}$ in Eq. (10)
21:     Update $\theta$ minimizing $\mathcal{L}_Q(\theta)$ in Eq. (1)
22:     Update $\phi$ minimizing $\mathcal{L}_\pi^{\text{adp,TD3+BC}}(\phi)$ in Eq. (11)
23:     Update $\psi, \varphi$ minimizing $\mathcal{L}_{\text{ELBO}}(s, a; \psi, \varphi)$ in Eq. (6)
24: **end for**

---

for high Q value as example. As TD3 utilizes the deterministic policy, we first generate a candidate action set $\mathcal{C} = \{a_i = \pi_\phi(s) + \epsilon_i\}_{i=1}^N$, where $\epsilon_i \sim \mathcal{N}(0, \delta)$ is the Gaussian noise. Then, candidate actions with top-$k$ Q values are selected as finalist action set $\mathcal{C}_f$. Finally, we sample the action for interaction from a categorical distribution $\mathrm{P}(a_i), a_i \in \mathcal{C}_f$ in Eq. (8) according to the uncertainty. In practice, we set $N = 100$ and $k = 10$ to interact with the environment by selecting actions that possess high value and relatively high uncertainty.

In terms of exploitation, SUNG (TD3+BC) follows TD3+BC and TD3 to take the standard TD learning for policy evaluation, and adopts the proposed adaptive exploitation method for policy improvement as

$$\mathcal{L}_\pi^{\text{adp,TD3+BC}}(\phi) = \mathbb{E}_{s \sim \mathcal{B}_{\text{OORB}}}\left[-Q_\theta\left(s, \pi_\phi\left(s\right)\right)\right] + \lambda \mathbb{E}_{(s,a) \sim \mathcal{B}_{\text{OORB}}}\left[\mathcal{I}\left(s, \pi_\phi\left(s\right)\right)\left(\pi_\phi\left(s\right) - a\right)^2\right].$$
(11)

Integrating the details above, we give a detailed algorithm description of SUNG (TD3+BC) in Algorithm 2.

## A.3 SUNG (CQL): BRIDGING CQL AND SAC

Then, we show how SUNG (CQL) bridges CQL and SAC to finetune the stochastic policy $\pi_\phi(\cdot|s)$ and two Q value functions $Q_{\theta_1}(s, a), Q_{\theta_2}(s, a)$.

In terms of exploration, the behavior policy of SUNG (CQL) is only compatible with preference for high uncertainty, as demonstrated in the main body. As SAC utilizes the stochastic policy, we first generate a candidate action set $\mathcal{C} = \{a_i \sim \pi_\phi(\cdot|s)\}_{i=1}^N$. Then, candidate actions with top-$k$ uncertainty are selected as finalist action set $\mathcal{C}_f$. Finally, we sample the action for interaction from a categorical distribution $\mathrm{P}(a_i), a_i \in \mathcal{C}_f$ similar to Eq. (8) according to the Q value. In practice, we set $N = 100$ and $k = 20$. Note that we slightly increase the value of $k$ to avoid the selected actions with too high uncertainty for sample efficiency. As such, we can interact with the environment by selecting actions that possess relatively high uncertainty and relatively high Q value.

---

**Algorithm 3** SUNG (CQL): Bridging CQL and SAC

---

1: **Input:** Dataset $\mathcal{D} = \{(s, a, r, s')\}$
2: // **Offline Pretraining**
3: Initialize Q network $Q_\theta$, policy network $\pi_\phi$, VAE network with $p_\psi$ and $q_\varphi$
4: **for** $t = 1$ **to** $T_1$ **do**
5:     Sample minibatch of transitions $(s, a) \sim \mathcal{D}$
6:     Update $\theta$ and $\phi$ minimizing $\mathcal{L}_Q^{\text{CQL}}(\theta)$ in Eq. (4) and $\mathcal{L}_\pi^{\text{SAC}}(\phi)$ in Eq. (12)
7:     Update $\psi, \varphi$ minimizing $\mathcal{L}_{\text{ELBO}}(s, a; \psi, \varphi)$ in Eq. (6)
8: **end for**
9: // **Online Finetuning**
10: Initialize OORB with $\mathcal{B}_{\text{OORB}} \leftarrow \{\mathcal{B} = \emptyset, \mathcal{D}\}$
11: **for** $t = 1$ **to** $T_2$ **do**
12:     // **Optimistic Exploration**
13:     Generate candidate action set $\mathcal{C} = \{a_i \sim \pi_\phi(\cdot|s)\}_{i=1}^N$
14:     Select actions with top-$k$ uncertainty as finalist action set $\mathcal{C}_f$
15:     Sample the action $a_t \sim \text{P}(a_i), a_i \in \mathcal{C}_f$ similar to Eq. (8) according to the Q value
16:     Execute action $a_t$, collect $s_{t+1}$ and $r_t$ from the environment
17:     Store transitions $(s_t, a_t, r_t, s_{t+1})$ into $\mathcal{B}$ and $\mathcal{D}$
18:     // **Adaptive Exploitation**
19:     Sample minibatch of transitions $(s, a, r, s') \sim \mathcal{B}_{\text{OORB}}$
20:     Construct the OOD identifier $\mathcal{I}$ in Eq. (10)
21:     Update $\theta$ minimizing $\mathcal{L}_Q^{\text{adp,CQL}}(\theta)$ in Eq. (13)
22:     Update $\phi$ minimizing $\mathcal{L}_\pi^{\text{SAC}}(\phi)$ in Eq. (12)
23:     Update $\psi, \varphi$ minimizing $\mathcal{L}_{\text{ELBO}}(s, a; \psi, \varphi)$ in Eq. (6)
24: **end for**

---

In terms of exploitation, SUNG (CQL) follows CQL and SAC to perform policy improvement as

$$\mathcal{L}_\pi^{\text{SAC}}(\phi) = \mathbb{E}_{s \sim \mathcal{B}_{\text{OORB}}, a \sim \pi_\phi(\cdot|s)} \left[ Q_\theta(s, a) - \log \pi_\phi(a|s) \right], \tag{12}$$

and adopts the proposed adaptive exploitation method for policy evaluation as

$$\mathcal{L}_Q^{\text{adp,CQL}}(\theta) = \mathbb{E}_{(s,a,r,s') \sim \mathcal{B}_{\text{OORB}}} \left[ \left( Q_\theta(s, a) - r - \gamma Q_{\bar{\theta}}(s', \pi_\phi(s')) \right)^2 \right]$$
$$+ \lambda \mathbb{E}_{s \sim \mathcal{B}_{\text{OORB}}} \left[ \mathcal{I}(s, \pi_\phi(s)) \left( \mathbb{E}_{a \sim \pi_\phi}[Q_\theta(s, a)] - \mathbb{E}_{a \sim \mathcal{B}_{\text{OORB}}}[Q_\theta(s, a)] \right) \right], \tag{13}$$

where $\pi_\phi(s)$ denotes the mean of the Gaussian stochastic policy. Integrating the details above, we give a detailed algorithm description of SUNG (CQL) in Algorithm 3.

## B    WHY EXPLORATION WITH PREFERENCE FOR HIGH VALUE FAILS WHEN FINETUNING CQL?

In empirical results, we find that O3F (Mark et al., 2022), which selects actions with preference for high Q value during exploration, fails when combined with CQL (Kumar et al., 2020). Besides, SUNG with preference of high Q value for exploration also fails when combined with CQL. Now, we give a qualitative analysis of this reason. To begin with, we recall the practical implementation of the policy evaluation in CQL:

$$\arg \min_Q \mathbb{E}_{s \in \mathcal{D}} \left[ \log \sum_a \exp(Q(s, a)) - \mathbb{E}_{a \sim \pi_\beta(a|s)}[Q(s, a)] \right] + \mathbb{E}_{(s,a,r,s') \sim \mathcal{D}} \left[ (Q(s, a) - y)^2 \right], \tag{14}$$

where $\pi_\beta$ denotes the behavior policy of the offline dataset and $y$ denotes the standard TD target for policy evaluation.

There are three terms in the above equation: given a state, the first term minimizes Q values of all the actions; the second term maximizes Q values of the action collected in the dataset; the third

term is the standard TD learning. As such, given state $s$, when we choose high-value action $a_h$ for exploration, the transition $(s, a_h, s', r)$ will be collected into the replay buffer. Then, during exploitation, the combination of the first and second term in the equation leads to a situation where most unseen actions have low value and will have lower value due to the first term, while few seen actions have high value and will have higher value due to the second term. As the third term involves bootstrapping, the high-value action will be updated to have a lower value, leading to a consistent decrease in the values of all actions with each gradient step. This can ultimately cause the collapse of the Q value functions, further resulting in poor policy through propagation in the policy improvement.

We remark that SUNG can solve this issue by altering the ranking criteria for the finalist action set and $k$ to show preference for high uncertainty while relatively high Q value during exploration.

## C    MORE EXPERIMENTAL DETAILS

### C.1    EXPERIMENTAL SETTING DETAILS

For MuJoCo tasks, we follow prior work (Zhang et al., 2023) to omit the evaluation on medium-expert and expert datasets, where offline pre-trained agents can already achieve expert-level performance and do not need further finetuning. Thus, we only focus on three kinds of quality: random (r), medium (m) and medium-replay (m-r).

For fair comparison, all the evaluations use datasets of the latest released "-v2" version that has fixed some potential bugs in the offline dataset. All the experiments are repeated with 5 different random seeds. For each evaluation, we run 10 episodes for MuJoCo tasks and 100 episodes for AntMaze tasks. We report the average D4RL score over last 3 evaluations for MuJoCo tasks, and last 10 evaluations for AntMaze tasks.

For offline pretraining, we first train the agents with 1M gradient steps by taking the suggested standard pipeline of TD3+BC and CQL:

- TD3+BC (Fujimoto & Gu, 2021): `https://github.com/sfujim/TD3_BC`
- CQL (Kumar et al., 2020): `https://github.com/young-geng/CQL`

Particularly, for AntMaze tasks, we strictly follow their original implementations to apply the reward transformation by subtracting 1 and 5 from all the rewards for SPOT and CQL, respectively.

Then, the last checkpoint of the pre-trained agents are used to warm start the online finetuning. Since the authors of SPOT (Wu et al., 2022) have provided the checkpoints of the pre-trained agents in `https://github.com/thuml/SPOT`, we directly adopt these for online finetuning. We conduct all the experiments on 2 NIVIDA 3090 GPUs and Intel Xeon Gold 6230 CPU at 2.10GHz.

### C.2    IMPLEMENTATION DETAILS AND HYPER-PARAMETERS

For VAE training, we follow the official implementation of BCQ (Fujimoto et al., 2019) and SPOT (Wu et al., 2022). Following SPOT, we normalize the states in the dataset for MuJoCo domains but do not normalize the states for AntMaze domains. Different from SPOT that utilizes VAE to estimate the density of the behavior policy, SUNG utilizes VAE to estimate the state-action visitation density. Thus, we set the latent dimension as $2 \times (\text{state dim} + \text{action dim})$ instead of $2 \times \text{action dim}$. Detailed hyper-parameters for VAE training are listed in Table 4. Note that we simply take the hyper-parameters reported in SPOT (Wu et al., 2022) and omit hyper-parameter tuning for VAE training. Accordingly, further performance improvement is expected from careful tuning.

For online finetuning, we follow the same standard architecture and the same hyper-parameter of TD3+BC, CQL, and SPOT for all the baselines. Note that we do not tune any hyper-parameter of backbone RL and architecture in SUNG for fair comparison. We present the hyper-parameters of SUNG for MuJoCo domains in Table 5. Specifically, for CQL-10, we proportionally reduce the number of the candidate action and finalist action to balance the computational costs brought by the ensemble technique. Besides, we reduce the OOD sample percentage to 5 because the ensemble technique may naturally eliminate the distribution shift issue (Lee et al., 2022).

Table 4: Hyper-parameters of VAE for uncertainty quantification in SUNG.

|  | Hyper-parameter | Value |
|---|---|---|
| VAE training | Optimizer | Adam |
|  | Learning rate | 1e-3 |
|  | Batch size | 256 |
|  | Gradient steps | 1e5 |
|  | KL term weight | 0.5 |
|  | State Normalization | True for MuJoCo |
|  |  | False for AntMaze |
| VAE architecture | Encoder hidden num | 750 |
|  | Encoder layers | 3 |
|  | Latent dim | $2 \times$ (state dim + action dim) |
|  | Decoder hidden dim | 750 |
|  | Decoder layers | 3 |

Table 5: Hyper-parameters of online finetuning in SUNG.

|  |  | TD3+BC | CQL | CQL-10 |
|---|---|---|---|---|
| SUNG | Candidate action num $N$ | 100 | 100 | 20 |
|  | Finalist action num $k$ | 10 | 20 | 4 |
|  | Softmax temperature $\alpha$ | | 1.0 | |
|  | OOD sample percentage $p$ | 5 | 10 | 5 |
| OORB | Online buffer size | | 2e4 | |
|  | Offline buffer size | | 2e6 | |
|  | OORB probability $p_{\mathrm{OORB}}$ | | 0.1 | |

In terms of baseline reproduction, we strictly follow their official implementation and hyper-parameters reported in their original paper to finetune agents pre-trained with TD3+BC, CQL and SPOT for MuJoCo and AntMaze domains. One exception is that the heavy ensemble technique is removed in BR for fair comparison. For fair comparison, we compare SUNG with the full implementation of (Lee et al., 2022) as BRPQ in Section 5.5. Note that we also equip the baseline with OORB to achieved improved performance for fair comparison.

## D    MORE EXPERIMENTAL RESULTS

In this section, we present supplementary experimental results to underscore the efficacy of SUNG and offer deeper insights. To begin with, we furnish results pertaining to the AntMaze domain. Subsequently, we conduct a comparative analysis with various other baselines, including IQL (Kostrikov et al., 2022), ODT (Zheng et al., 2022), ACA (Yu & Zhang, 2023), and E2O (Zhao et al., 2023a). Moreover, we include comprehensive learning curves to complement the numerical findings. Lastly, we incorporate additional ablation studies and hyperparameter analyses to enhance comprehension of SUNG.

### D.1    EXPERIMENTS ON ANTMAZE

We conduct experiments on more challenging AntMaze domains, which require the agents to deal with the sparse rewards and stitch fragments of sub-optimal trajectories. Results for AntMaze domains with SPOT and CQL as backbone offline RL method are shown in Table 6 and Table 7, respectively. From the results, we can observe that online-ft, BR and PEX struggle to benefit from online finetuning in most settings, except for the relatively easy antmaze-umaze task. This stems from the need for explicit or implicit behavior cloning that is offered by offline RL objectives, which is required to handle the sparse rewards and stitch sub-optimal sub-trajectories. However, online-ft, BR and PEX are developed on top of purely online RL methods, thus failing in most of the tasks.

Table 6: Comparison of the averaged D4RL score on AntMaze tasks with **SPOT** as the offline RL backbone method. We report the mean and standard deviation over 5 seeds.

| | SPOT | offline-ft | online-ft | BR | O3F | APL | PEX | PROTO | SUNG |
|---|---|---|---|---|---|---|---|---|---|
| antmaze-umaze-v2 | 93.2 | 96.6±0.4 | 88.3±2.3 | 92.9±0.5 | 96.9±0.6 | **97.6±1.0** | 12.2±13.0 | 91.7±1.9 | 97.0±0.8 |
| antmaze-umaze-diverse-v2 | 41.6 | 22.0±18.1 | 0.0±0.0 | 0.0±0.0 | 65.3±4.4 | 50.5±30.7 | 0.0±0.0 | 0.0±0.0 | **71.7±3.6** |
| antmaze-medium-play-v2 | 75.2 | 83.9±2.2 | 0.0±0.0 | 6.4±9.3 | 85.6±2.4 | 86.0±2.3 | 0.1±0.1 | 5.5±10.6 | **88.6±2.9** |
| antmaze-medium-diverse-v2 | 73.0 | 84.6±1.1 | 0.5±0.8 | 0.0±0.0 | 80.1±7.8 | 86.0±4.4 | 0.0±0.0 | 0.0±0.0 | **91.7±1.6** |
| antmaze-large-play-v2 | 40.8 | 40.9±4.6 | 0.0±0.0 | 0.0±0.0 | 38.8±11.2 | 38.9±4.7 | 0.0±0.0 | 0.0±0.0 | **45.7±4.5** |
| antmaze-large-diverse-v2 | 44.0 | 16.6±2.2 | 0.0±0.0 | 0.0±0.0 | 3.0±0.8 | 3.8±1.6 | 0.0±0.0 | 0.0±0.0 | **19.8±15.8** |
| **Total** | 367.8 | 344.6 | 88.8 | 99.3 | 369.7 | 362.8 | 12.3 | 97.3 | **414.5** |

Table 7: Comparison of the averaged D4RL score on AntMaze tasks with **CQL** as the offline RL backbone method. We report the mean and standard deviation over 5 seeds.

| | CQL | offline-ft | online-ft | BR | O3F | APL | PEX | PROTO | SUNG |
|---|---|---|---|---|---|---|---|---|---|
| antmaze-umaze-v2 | 88.6 | 87.9±3.3 | 96.0±1.3 | 68.5±38.1 | **98.6±0.3** | 96.0±1.3 | 87.3±5.2 | 97.6±1.0 | 96.9±0.7 |
| antmaze-umaze-diverse-v2 | 41.0 | 45.9±4.3 | 0.0±0.0 | 0.0±0.0 | 0.0±0.0 | 0.0±0.0 | 0.7±0.9 | 0.0±0.0 | **50.5±9.6** |
| antmaze-medium-play-v2 | 63.8 | 75.4±3.5 | 18.6±15.3 | 22.2±25.7 | 67.3±17.0 | 22.8±28.2 | 0.3±0.3 | 1.4±2.6 | **86.3±2.1** |
| antmaze-medium-diverse-v2 | 61.8 | 74.1±2.0 | 22.1±25.6 | 13.6±20.8 | 80.8±4.9 | 36.8±33.4 | 0.3±0.6 | 1.1±2.2 | **85.6±4.5** |
| antmaze-large-play-v2 | 32.0 | 41.8±8.0 | 0.1±0.2 | 0.0±0.0 | 0.0±0.0 | 0.0±0.0 | 0.0±0.0 | 0.0±0.0 | **52.7±9.8** |
| antmaze-large-diverse-v2 | 32.6 | 35.3±10.7 | 0.7±1.4 | 0.1±0.1 | 0.0±0.0 | 0.0±0.0 | 0.0±0.0 | 0.0±0.0 | **44.1±12.3** |
| **Total** | 319.6 | 360.4 | 137.5 | 104.3 | 246.7 | 155.6 | 88.6 | 100.1 | **416.1** |

Furthermore, we remark that SUNG, when combined with either SPOT or CQL, outperforms other state-of-the-art methods in most settings. This demonstrates the superiority of SUNG for discovering informative state-action pairs that can lead to high outcomes, thereby providing valuable behavior patterns for agents to learn from. Moreover, SUNG achieves flexible trade-off between offline and online RL objectives by tuning $p$, which enables stitching sub-optimal sub-trajectories and overcoming the sparse reward issue.

## D.2 COMPARISON WITH OTHER OFFLINE-TO-ONLINE RL METHODS

In this subsection, we compare SUNG with other offline-to-online RL methods that are designed for one specific offline RL method. Concretely, we select the following baselines: IQL (Kostrikov et al., 2022), ODT (Zheng et al., 2022) and ACA (Yu & Zhang, 2023). We follow the original experimental setting in ODT to take 200K environment steps and only consider tasks with medium and medium-replay offline datasets for online finetuning to achieve fair comparison. The results are directly copied from (Yu & Zhang, 2023).

As shown in Table 8, for almost all the settings, SUNG when combined with either TD3+BC or CQL outperforms other methods by a large margin in both final performance and online improvement $\delta$. IQL presents conservative online finetuning, which can stably improve the performance but derive limited performance under a limited interaction setting, i.e., 200K environment steps. ODT learns a reward-conditioned policy in a supervised learning paradigm, thereby struggling to fully benefit from the trail-and-error paradigm used in traditional online RL. However, it is worth noting that ODT enables users to customize the behavior of the agents by specifying different conditions, which is not featured by traditional RL.

## D.3 COMPARISON WITH E2O

In this subsection, we compare SUNG with E2O (Zhao et al., 2023a), a most recent ensemble-based offline-to-online RL method. We strictly follow E2O to focus on medium-level offline datasets. Here, we offer two variants of SUNG: **SUNG w/ VAE** and **SUNG w/ Var**. These variants differ in their uncertainty quantification methods. The former employs a VAE for state-action density estimation, while the latter uses the variance of 10 Q functions to estimate epistemic uncertainty. As demonstrated in Table 9, SUNG achieves competitive experimental results when compared to

Table 8: Comparison of the averaged D4RL score on MuJoCo tasks, with a focus on both final performance and online improvement $\delta$. m = medium, m-r = medium-replay. We run **200K** environment steps to keep consistent with ODT's setting. We report the mean value over 5 seeds.

| | IQL | $\delta$ | ODT | $\delta$ | ACA | $\delta$ | SUNG (TD3+BC) | $\delta$ | SUNG (CQL) | $\delta$ |
|---|---|---|---|---|---|---|---|---|---|---|
| **halfcheetah-m-v2** | 47.4 | 0.0 | 42.1 | -0.2 | 72.7 | 26.3 | **85.5** | **37.5** | **86.7** | **38.6** |
| **hopper-m-v2** | 66.8 | 3.0 | 89.8 | 27.4 | 99.3 | **42.4** | 97.1 | 35.7 | **106.1** | 32.4 |
| **walker2d-m-v2** | 80.3 | 0.4 | 73.7 | 1.6 | 76.1 | -3.3 | **115.4** | 33.2 | 114.6 | 30.3 |
| **halfcheetah-m-r-v2** | 44.1 | 0.0 | 40.6 | 3.0 | 64.3 | 22.1 | **79.8** | 35.1 | **82.9** | 36.0 |
| **hopper-m-r-v2** | 96.2 | 4.1 | 88.3 | 5.1 | 103.2 | **53.9** | **104.1** | 48.2 | 83.5 | -12.5 |
| **walker2d-m-r-v2** | 70.6 | -3.1 | 73.1 | 5.2 | 82.1 | 18.9 | **110.7** | **39.0** | 116.7 | 33.3 |
| **Total** | 405.5 | 4.5 | 407.6 | 42.1 | 497.6 | 160.3 | **592.6** | **228.7** | 590.5 | 158.1 |

Table 9: Comparison of the averaged D4RL score on MuJoCo tasks with CQL-10 as the offline RL backbone method. We report the mean and standard deviation over 5 seeds.

| | CQL-10 | E2O | SUNG w/ VAE | SUNG w/ Var |
|---|---|---|---|---|
| **halfcheetah-m-v2** | 55.0 | **99.7**±**1.3** | 96.6±1.0 | 98.8±1.3 |
| **hopper-m-v2** | 66.9 | 109.8±1.9 | 111.4±0.8 | **112.5**±**1.1** |
| **walker2d-m-v2** | 83.2 | 104.9±16.2 | **113.8**±**2.6** | 110.2±9.5 |
| **halfcheetah-m-r-v2** | 52.6 | 94.4±1.4 | 92.5±0.5 | **96.6**±**1.0** |
| **hopper-m-r-v2** | 102.3 | **107.9**±**6.7** | 100.7±13.6 | 107.4±4.8 |
| **walker2d-m-r-v2** | 82.1 | **122.6**±**11.1** | 105.1±14.6 | 113.3±16.8 |
| **Total** | 501.0 | **639.3** | 620.0 | 638.8 |

E2O, a method specifically designed for ensemble-based offline-to-online RL. This showcases the effectiveness of our proposed framework. Furthermore, it is worth noting that our uncertainty quantification method can attain comparable performance to using the variance of 10 Q functions, which would significantly increase computational costs.

## D.4 LEARNING CURVES

We display the learning curves of SUNG when combined with TD3+BC and CQL for MuJoCo domains in Fig. 5. Besides, we display the learning curves of SUNG when combined with SPOT and CQL for AntMaze domains in Fig. 6. In most settings, we can observe that SUNG surpasses the previous state-of-the-art methods in terms of both final performance and sample efficiency.

## D.5 ABLATION STUDY

**Ablation Study with Variants of SUNG (Fig. 2).** We consider the following variants of SUNG to perform the ablation study over the components of our framework:

- **w/o Opt. Exploration:** We replace the proposed optimistic exploration strategy by the default exploration provided by TD3 and SAC. For TD3, we select action with Gaussian noise for exploration, i.e., $\pi_E(s) = \pi_\phi(s) + \epsilon, \epsilon \sim \mathcal{N}(0, \delta)$. For SAC, we sample action from the policy for exploration, i.e., $\pi_E(s) = a, a \sim \pi_\phi(\cdot|s)$.

- **Opt. Exploration w/o Unc.:** We remove the uncertainty from the proposed exploration strategy to greedily select action that maximizes Q value, i.e., $\pi_E(s) = \arg\max_{a \in \mathcal{C}} Q(s, a)$.

- **Opt. Exploration w/o Q:** We remove the Q value from the proposed exploration strategy to greedily select action that maximizes the uncertainty, i.e., $\pi_E(s) = \arg\max_{a \in \mathcal{C}} \mathcal{U}(s, a)$.

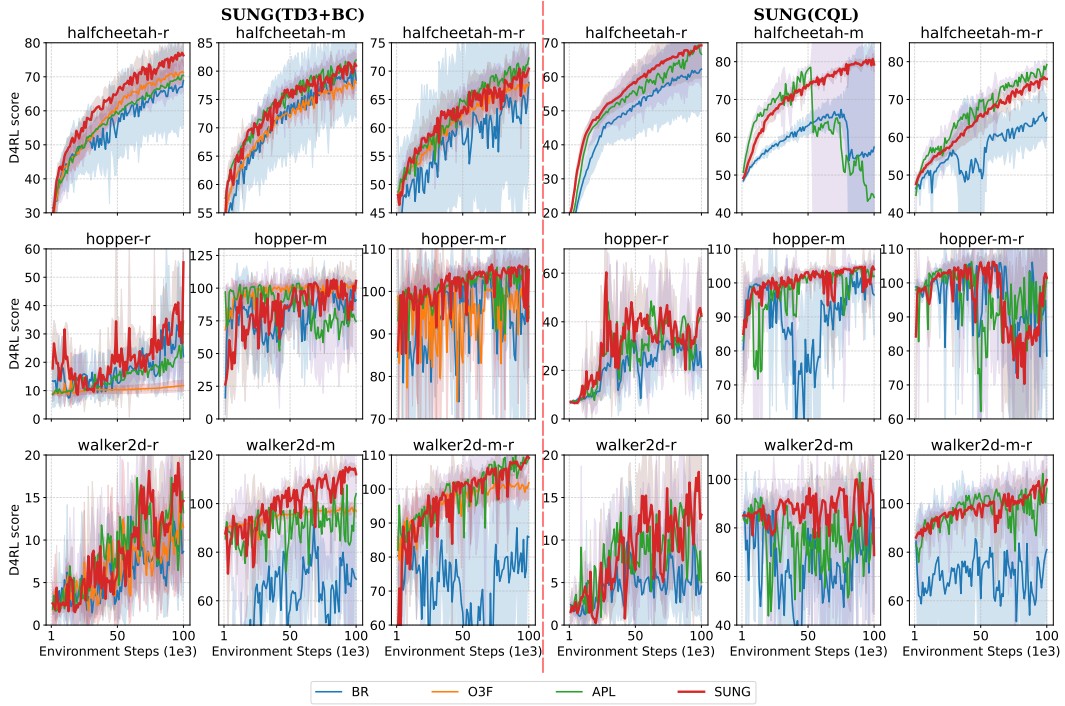

Figure 5: Learning curves of SUNG, when combined with TD3+BC and CQL, for 100K environment steps on **MuJoCo** tasks. We also include three best-performing baselines, BR, O3F and APL, for comparison. w = Walker, r = random, m = medium, m-r = medium-replay. We report the mean D4RL score and standard deviation over 5 training seeds with 10 evaluation episodes each. Note that O3F is omitted when combined with CQL due to its divergence.

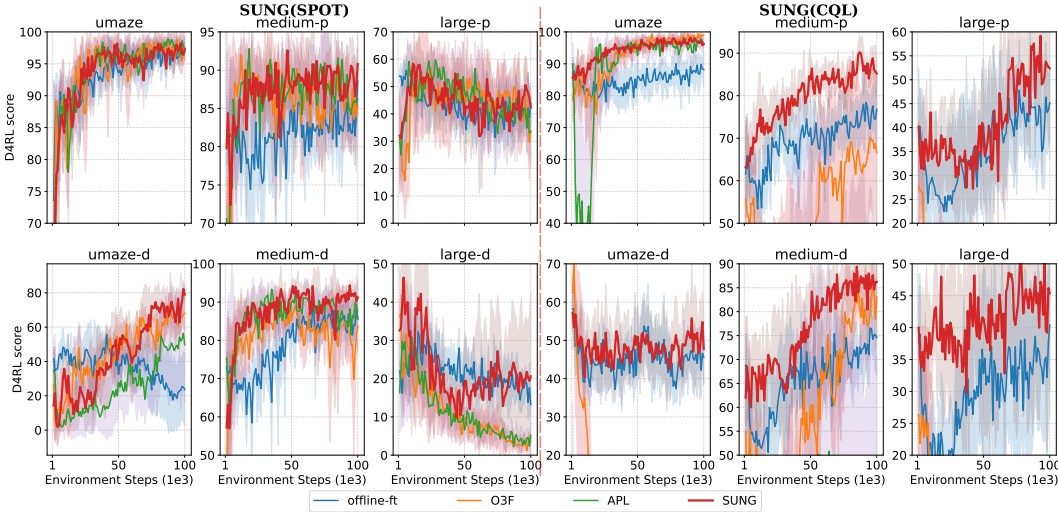

Figure 6: Learning curves of SUNG, when combined with SPOT and CQL, for 100K environment steps on **AntMaze** tasks. We also include three best-performing baselines, offline-ft, O3F and APL, for comparison. p = play, d = diverse. We report the mean D4RL score and standard deviation over 5 training seeds with 100 evaluation episodes each.

- **w/o Adp. Exploitation:** We replace the proposed adaptive exploitation method by the standard online RL objective of TD3 and SAC, respectively.

- **Unc. Estimation w/ Q std:** We replace VAE by the standard deviation of two Q value functions for uncertainty estimation, i.e.,

$$\mathcal{U}(s,a) = \sigma(Q_{\theta_1}(s,a), Q_{\theta_2}(s,a)) = |Q_{\theta_1}(s,a) - Q_{\theta_2}(s,a)|.$$

Besides, we also consider two more ablation settings:

- **Adp. Exploitation w/ Rand.:** We construct the OOD state-action pair set $\mathcal{D}_{\text{OOD}}$ by randomly sampling from the mini-batch instead of using the proposed OOD sample identifier.

- **OORB w/o Offline Data:** We do not utilize transitions from offline dataset for online finetuning by removing offline data from OORB.

We present the results of the two ablation studies above in Fig. 7 and Fig. 8, respectively. As shown in Fig. 7, we can find when we use a random sampling strategy for OOD sample identification, the performance on most settings deteriorates. As shown in Fig. 8, we observe that the removal of offline data significantly hurts the offline-to-online performance. This is because online finetuning can benefit from reusing the diverse behaviours contained in the offline dataset to avoid overfitting.

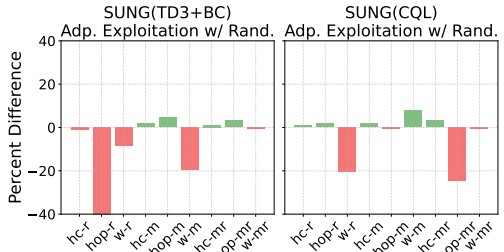

Figure 7: Performance difference of the performance of random strategy for OOD sample identification, compared with the full algorithm.

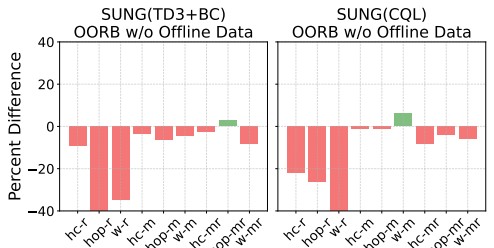

Figure 8: Performance difference of the performance without offline data, compared with the full algorithm.

**Hyper-parameter Analysis on Softmax Temperature** $\alpha$. Here, we give a hyper-parameter analysis of the softmax temperature $\alpha$ in Eq. (8). Specifically, we investigate the choice of $\alpha \in \{0.1, 0.2, 0.5, 1.0, 5.0, 10.0, 20.0, 100.0\}$ for MuJoCo domains in Fig. 9. As shown in the figure, we can find that $\alpha$ is a relatively insensitive hyper-parameter for the offline-to-online performance, and that the best-performing choice of $\alpha$ varies among different settings. Thus, though a careful tuning on $\alpha$ will bring more offline-to-online improvement, we set $\alpha = 1.0$ in all the experimental result throughout the paper to guarantee the simplicity of the proposed framework.

**Hyper-parameter Analysis on OORB Sampling Probability** $p_{\text{OORB}}$. Furthermore, we give a hyper-parameter analysis on $p_{\text{OORB}}$ for the simple-yet-effective OORB. Specifically, we investigate the choice of OORB sampling probability $p_{\text{OORB}} \in \{0.0, 0.1, 0.2, 0.4, 0.6, 0.8, 1.0\}$ for MuJoCo domains in Fig. 10. Note that a higher probability $p_{\text{OORB}}$ indicates more reuse of the offline dataset during policy learning. From the results, we can observe that $p_{\text{OORB}} = 0.1$ performs the best when combined with either TD3+BC and CQL, while either setting $p_{\text{OORB}}$ too conservatively (i.e., $p_{\text{OORB}} = 0$) or too adversely (i.e., $p_{\text{OORB}} = 1.0$) hurts the performance. As pointed out by Zheng et al. (Zheng et al., 2023), offline data can prevent agents from prematurely converging to sub-optimal policies due to the potential data diversity, while online data can stabilize training and accelerate convergence. Thus, it is crucial for sample-efficient offline-to-online RL algorithms to incorporate both offline and online data during policy learning.

### D.6 Decay on Regularization Weight $\lambda$

An intuition for the proposed adaptive exploitation method is that the regularization weight $\lambda$ in Eq. (9) should be decayed during online finetuning, since the state-action distribution shift gradually eliminates with the environment step. Hence, in this subsection, we explore the decay mechanism

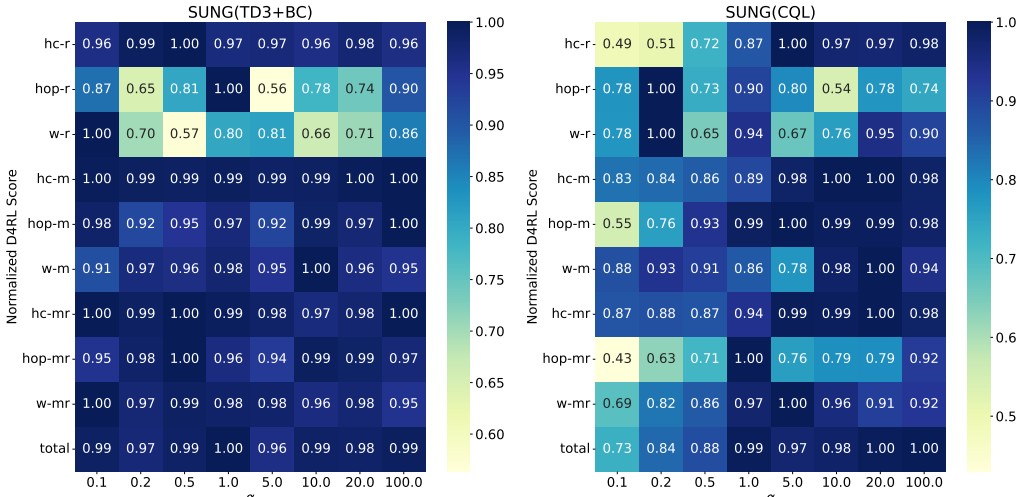

Figure 9: Heatmap for performance of SUNG with different softmax temperature $\alpha$. The results of each setting are normalized with maximum normalization for better visualization. hc = HalfCheetah, hop = Hopper, w = Walker, r = random, m = medium, mr = medium-replay.

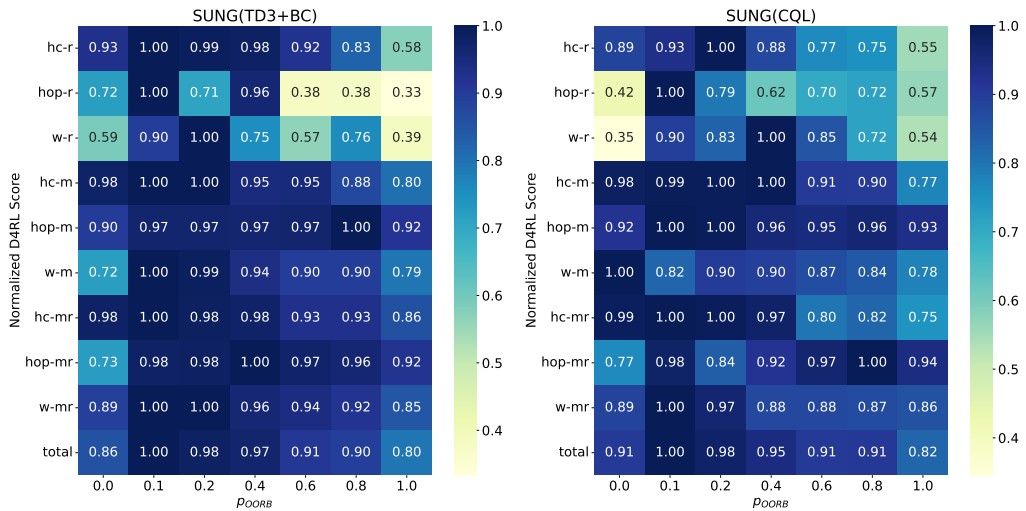

Figure 10: Heatmap for performance of SUNG with different OORB sampling probability $p_{\text{OORB}}$. The results of each setting are normalized with maximum normalization for better visualization. hc = HalfCheetah, hop = Hopper, w = Walker, r = random, m = medium, mr = medium-replay.

for $\lambda$ by linearly decreasing $\lambda$ to $\{0.0, 0.2, 0.4, 0.6, 0.8, 1.0\}$ times its initial value at the end of online finetuning. As shown in Fig. 11, no decay mechanism outperforms the other in terms of average total performance when combined with either TD3+BC or CQL. We conjecture that this is because 100K environment steps are not enough for agents to fully overcome the distribution shift issue in most settings. We also note that no decay mechanism does not consistently outperform all the other settings, which depends on the quality of the dataset and the difficulty of the task. Hence, careful tuning of the decay mechanism may bring extra offline-to-online improvement, especially for a larger amount of online environment steps. In this work, we do not utilize the decay mechanism for simplicity. Moreover, we also remark that SUNG with different decay mechanism still outperforms the best-performing baseline, which further demonstrates the superiority of the proposed framework.

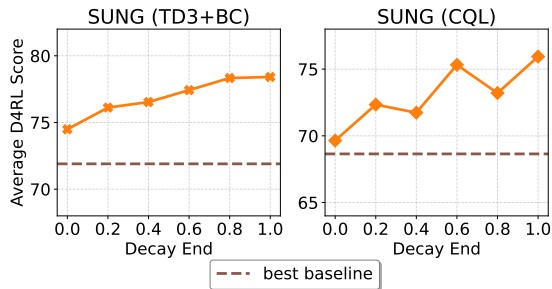

Figure 11: Performance comparison of SUNG with different decay end for regularization weight $\lambda$, where $x$ axis means decay until reaching $x \cdot \lambda$ at the end of online finetuning. We present the D4RL score averaged over MuJoCo domains. $x = 1.0$ represents SUNG without decay mechanism.

## E   LIMITATION, FUTURE WORK AND BROADER IMPACTS

While achieving promising performance, the proposed framework also has some inherent limitations. Some recently developed offline RL methods can be built on top of on-policy RL (Zhuang et al., 2023) and supervised learning paradigm (Chen et al., 2021; Emmons et al., 2022; Xu et al., 2022) instead of pure off-policy RL. Hence, one limitation is that SUNG is not compatible with agents pretrained by these new paradigms for offline RL. However, we argue that the insights of SUNG can be applied to them in principle. We leave the exploration of its implementation for combining with on-policy RL and supervised learning as an interesting future work. In the current version, SUNG investigates the offline-to-online RL setting to solve tasks whose inputs are the states of the simulated environment. It would be interesting to see whether SUNG is also superior with the visual RL setting (Lu et al., 2022). Moreover, further investigation on uncertainty estimation, replay buffer, and more backbone offline RL methods needs to be explored. Finally, we remark that the exploration of its application on real-world tasks is also an interesting future work.

We hope our work showcases the potential of the offline pretraining and online finetuning paradigm in effectively and efficiently deploying RL algorithms in real-world applications. Besides, we hope that our work inspires future research to develop more sample-efficient offline-to-online RL methods. We do not see negative societal impacts of our work in practice.

