# OpenReview forum: "A Simple Unified Uncertainty-Guided Framework for Offline-to-Online Reinforcement Learning"
_ICLR.cc/2024/Conference — Submitted to ICLR 2024_

### Official Review · Reviewer_MwS2 · 2023-10-29

**Soundness:** 2 fair
**Presentation:** 2 fair
**Contribution:** 2 fair
**Rating:** 3
**Confidence:** 5

**Summary:**

This work investigates fine-tuning pretrained offline RL policies via further online interactions, aiming to address two challenges: constrained exploratory behavior and state-action distribution shift. Building on uncertainty quantification, this work introduces a
fine-tuning framework that alternates between optimistic exploration and adaptive exploitation.

**Strengths:**

+ Overall, the paper is well written and easy to read. The challenges and motivation are clearly phrased.

+ The studied problem, offline-to-online RL, is an important problem,  especially when fine-tuning large-scale policy models.

+ The proposed methods introduce some new ideas (e.g., uncertainty-weighted online exploitation) capable of benefiting the development of this field.

**Weaknesses:**

One main weakness of this work mainly lies in (i) the explainability and usability of the proposed methods, and (ii) the gap between the motivation and evaluation (please see the details below).

- This work utilizes VAE to characterize the uncertainty of state-actions, whereas the paper does not provide the specific definition of uncertainty, and it’s not convincing why VAE is superior to other methods, e.g., Q-ensemble learning.

- The reviewer is concerned with the highlighted OOD (distributional shift) issue. In Section 4.3, the paper claims that the OOD state-actions can harm the performance of online fine-tuning. Wouldn’t the collected online data, that contain reward signals, enhance the data support of the offline dataset?

- There is a lack of clear explanation towards the unique challenges in the exploration-exploitation of the offline-to-online problem. For instance, a major issue could be the “forgetting” - during the model updates  the fine-tuned policy can quickly forget what it has learned from offline, which is neglected by this work.

- The motivation and evaluation are isolated. In the Introduction, the paper claims “the proposed method provides a generic solution for offline-to-online RL to enable finetuning agents pretrained with different offline RL objectives.” However, in the experiment, the tuning policy is pretrained from the same task.

**Questions:**

There are a set of hyper-parameters introduced in the proposed method without reasonable guidelines on how to determine these parameters, e.g., N, alpha, lambda, etc. It would hinder the usability of the algorithm, and it would help if the authors can provided theoretical guidance on how to select the hyper-parameters to optimally balance the exploration-exploitation tradeoff.

- As shown in Eq. (9), is the fine-tuning method dependent on the pretraining algorithm?

---

> ### Author Response · Authors · 2023-11-11
>
> Many thanks to Reviewer MwS2 for providing the detailed and valuable review, and we respond to the weaknesses and questions as below.
>
> > **Q1: Definition of uncertainty.**
>
> In this work, we define the uncertainty as the approximate negative log likelihood of the state-action density as shown in Page 4:
> $\mathcal{U}(s,a)\overset{\rm{def}}{=}-\log p(s,a)\approx\mathcal{L}_{\mathrm{ELBO}}(s,a;\psi,\varphi).$
>
> > **Q2: Advantage of VAE over ensemble-based technique for uncertainty quantification.**
>
> We detail the reasons at the beginning of Section 4.1 as below.
>
> > Many prior works typically utilize Q ensembles to qualify uncertainty. However, the ensemble technique may significantly increase computational costs. Thus, in this work, we utilize a simple yet effective approach by adopting VAE  as a state-action visitation density estimator for uncertainty quantification.
>
> Furthermore, we also derive some initial empirical results on the ensemble setting in Table 9 in the appendix. From Table 9, we can observe that uncertainty quantification with VAE can attain comparable performance to using the variance of 10 Q functions, where the latter would significantly increase computational costs.
>
> > **Q3: Enhance data support of the offline dataset through online interaction.**
>
> Yes, the collected online data can enhanced the data support of the offline dataset. However, we aim at 100k environment step setting in this work, which requires extremely high sample efficiency. Therefore, although the policy can automatically recover the extrapolation error brought by OOD state-actions via trial-and-error, it leads to inferior sample efficiency. Thus, we introduce the adaptive exploitation method with OOD sample identification to overcome this issue.
>
> > **Q4: Explanations on exploration-exploitation issue in offline-to-online RL.**
>
> We observe that aggressive exploration may exacerbate the distribution shift, while conservative exploitation can hinder agents from efficient online finetuning. Specifically, as stated at the end of Section 4.2, to solve the constrained exploratory behavior issue, we introduce optimistic exploration strategy. However, this may also bring negative effects by further increasing state-action distribution shift, as a result of the principle of optimism in the face of uncertainty. Thus, simultaneously and efficiently addressing both challenges brings additional improvement.
>
> > **Q5: "Forgetting" issue.**
>
> We want to clarify that the forgetting issue is one outcome of failing to handle state-action distribution shift. As stated in Introduction,
>
> > Accordingly, the state-action distribution shift between offline and online data occurs, leading to the well-known extrapolation error during exploitation, which may wipe out the good initialization obtained from the pretraining stage.
>
> Thus, we solve this issue by proposing the adaptive exploitation method with OOD sample identification.
>
> > **Q6: Evaluation of "generic" solution for offline-to-online RL.**
>
> We want to clarify that the genericness of SUNG means that SUNG can be seamlessly combined with multiple existing offline RL methods. We verify this by combing SUNG with four backbone offline RL methods, including TD3+BC, CQL, CQL-10, and SPOT.
>
> > **Q7: Hyper-parameter Tuning.**
>
> We want to clarify that we only tune these two hyper-parameters of SUNG in this paper, i.e., finalist action set size k and OOD sample percent p. Actually, SUNG does not require extensive hyper-parameter tuning; instead, it only demands minimal effort in that regard. As demonstrated in Figure 3, SUNG consistently outperforms the best baseline for any choice of k. Similarly, as illustrated in Figure 4, SUNG consistently outperforms the best baseline when p is appropriately set within the range of [5, 10]. Thus, SUNG achieves robust performance without the need for intensive hyper-parameter fine-tuning, which aligns with our initial motivation to develop a simple-yet-effective offline-to-online RL algorithm.
>
> In terms of other hyper-parameters, we set them with an appropriate value and never tune them out of simplicity. For example, we set N=100 for TD3+BC and CQL, and set them to 20 for CQL-10 to decrease the computational costs. We set alpha=1.0, which is the exactly the default setting without temperature. We set lambda=1.0, which is exactly the default multi-task loss setting. Thus, we also highlight that carefully tuning these hyper-parameters may bring further improvement of SUNG.
>
> > **Q8: Relationship between finetuning method and pretraining algorithm.**
>
> Yes, you are right. In this work, we utilize the corresponding online RL objectives for online finetuning to keep consistent.
>
> ---
> We hope that our response could address your concerns and that you would consider increasing your score.

---

> > ### Comment · Reviewer_MwS2 · 2023-11-21
> > **The answers to Q3, Q4 and A5 are not so convincing.**
> >
> > Appreciate the authors for the response.  I find the answers to Q3, Q4 and A5 are not so convincing.
> >
> > Specifically, for Q3, it is not clear how the adaptive exploration algorithm can achieve "extremely high sample efficiency" and do much better than the online data collection.
> >
> > For Q4, my question is about how to strike the balance between aggressive and conservative exploration.
> >
> > For Q5, how does the adaptive exploitation method with OOD sample identification overcome "forgetting"?

---

> ### Author Response · Authors · 2023-11-21
>
> Dear reviewer MwS2, as the deadline for author-reviewer discussions is approaching, could you kindly confirm if your concerns have been adequately addressed? We are more than willing to continue the discussion if there are any lingering issues. Thank you sincerely for your efforts!

---

### Official Review · Reviewer_YGBs · 2023-10-30

**Soundness:** 2 fair
**Presentation:** 3 good
**Contribution:** 2 fair
**Rating:** 3
**Confidence:** 4

**Summary:**

The paper presents the Simple Unified Uncertainty-Guided (SUNG) framework as a solution to the challenges encountered in offline-to-online reinforcement learning (RL). To address issues related to exploratory behavior and state-action distribution shifts, the framework leverages a VAE-based state-action visitation density estimator to quantify uncertainty. It also employs an optimistic exploration strategy to select actions with both high value and uncertainty, facilitating efficient exploration. Furthermore, SUNG incorporates an adaptive exploitation method that applies conservative offline RL objectives to high-uncertainty samples and standard online RL objectives to low-uncertainty samples, enabling a smooth transition from offline to online stages.

**Strengths:**

Research into the domain of online finetuning holds significant importance within the field of offline learning.

The experimental evaluation suggests that there is potential for improvement in the finetuning performance when the proposed approach is combined with various offline RL methods across a range of environments and datasets from the D4RL benchmark. These findings indicate the adaptability and practicality of the suggested technique in different settings.

The paper demonstrates a high degree of clarity and well-structured writing, rendering it easily understandable for readers.

**Weaknesses:**

The primary concern raised with regard to this paper pertains to its novelty. The concept of leveraging uncertainty in the context of offline learning is a well-established one. From the perspective of reviewers, the key innovation in this article lies in the utilization of a VAE for quantifying uncertainty, which does not represent a notable departure from conventional methods.

While this paper introduces a straightforward empirical method, it is notable for its absence of a comprehensive theoretical analysis to substantiate the advantages of the proposed approach relative to existing methods. Offering a theoretical foundation and analysis would be valuable in bolstering the method's credibility and potential impact, providing a more solid basis for its effectiveness.

**Questions:**

Plese see comments on weakness.

---

> ### Author Response · Authors · 2023-11-11
>
> **(Part 1/2)**
>
> ---
>
> We extend our sincere gratitude to Reviewer YGBs for valuable and constructive feedback. In response to the identified weaknesses and questions, we provide the following clarifications and improvements
>
> > **Q1: Novelty Concern.**
>
> We want to highlight our novelty from the following aspects.
>
> **Firstly**, we mark that we identify the key question in offline-to-online RL as simultaneously addressing both challenges of constrained exploratory behavior and state-action distribution shift while eliminating the inherent conflict between exploration and exploitation. While prior research has attempted to address each challenge independently, tackling both challenges simultaneously is not a trivial task. For instance, some previous works (e.g., O3F) proposes efficient exploration strategy to mitigate the constrained exploratory behavior. However, we observe their divergence and sub-optimal performance when finetuning CQL agents, precisely due to this inherent conflict. Please refer to Appendix B for detailed analyses. In contrast, SUNG naturally unifies the solutions to both challenges with the tool of uncertainty, achieving the desired trade-off between exploration and exploitation. As a result, when combined with either TD3+BC or CQL, SUNG significantly surpasses previous SOTAs.
>
> **Secondly**, we point out that we develop novel techniques, such as bi-level action selection mechanism and adaptive exploitation with OOD sample identification, enabling SUNG to present a simple-yet-effective manner. Different from previous optimistic exploration techniques, we propose a bi-level action selection mechanism to mitigate the challenges of computation complexity and intensive task-specific hyper-parameter tuning, while achieving efficient exploration. In contrast to previous adaptive exploitation methods, SUNG builds on the insight that offline RL objectives can be typically interpreted as an online RL objective and a regularization term. As such, we develop a novel adaptive exploitation method with OOD sample identification to achieve the desired trade-off between performance and stability. Significantly, SUNG demonstrates robust superiority over previous SOTAs across a diverse set of hyper-parameters, as shown in Figure 3 and 4, which further enhances its practical applicability.
>
> **Thirdly**, we emphasize that we perform systematic empirical evaluations for SUNG. We combine SUNG with 4 backbone offline RL algorithms across 15 settings spanning MuJoCo and AntMaze domains, and we utilize 13 baselines for comparison. Experimental results demonstrate that SUNG significantly outperforms previous SOTAs. Furthermore, we conduct extensive ablation studies on each component to thoroughly verify the effectiveness of SUNG. Additionally, we also showcase SUNG's seamless integration with other techniques, such as ensemble Q learning and high update-to-data ratios, leading to superior finetuning performance.
>
> **Finally**, we also contribute a series of fresh insights to the offline-to-online setting, e.g., (1) Exploration with high Q-value actions leads to divergence when finetuning CQL, as elaborated in Appendix B. SUNG can stably finetune CQL agents by balancing the trade-off between uncertainty and Q value. (2) Fully online RL objectives fail in challenging sparse-reward tasks, as detailed in Appendix D.1. SUNG can achieve flexible trade-off between offline and online RL objectives, which enables stiching sub-optimal sub-trajectories and overcoming the sparse reward issue.
>
> **Overall**, in this paper, we aim to develop a simple, generic, sample-efficient offline-to-online RL framework as a good starting point for deriving strong empirical performance.

---

> ### Author Response · Authors · 2023-11-11
>
> **(Part 2/2)**
>
> ---
>
> > **Q2: Concern on theoretical analysis.**
>
> We would like to clarify that the scope of this paper is to develop a generic offline-to-online RL framework that can be applied to different kinds of offline RL algorithms. Despite the practical theoretical results achieved by the offline RL community, to the best of my knowledge, there still exists a gap in establishing an initial theoretical foundation for offline-to-online RL. Moreover, given the diversity of theoretical foundations of offline RL algorithms, proposing a single framework considering different theoretical guarantees presents challenges. Therefore, we follow previous works [1-5] to focus on developing empirically effective and robust offline-to-online RL algorithms. From our systematic empirical evaluation on MuJoCo and AntMaze domains, SUNG significantly outperforms the previous SOTAs and does not require intensive hyper-parameter tuning. Moreover, extensive ablation studies on each component also demonstrate the effectiveness of SUNG.
>
> [1] Offline-to-Online Reinforcement Learning via Balanced Replay and Pessimistic Q-Ensemble, CoRL'22.
>
> [2] Online Decision Transformer, ICML'22.
>
> [3] Policy Expansion for Bridging Offline-to-Online Reinforcement Learning, ICLR'23.
>
> [4] Adaptive Policy Learning for Offline-to-Online Reinforcement Learning, AAAI'23.
>
> [5] Actor-Critic Alignment for Offline-to-Online Reinforcement Learning, ICML'23.
>
> ---
> We hope that our response could address your concerns and that you would consider increasing your score.

---

> ### Author Response · Authors · 2023-11-21
>
> Dear reviewer YGBs, as the deadline for author-reviewer discussions is approaching, could you kindly confirm if your concerns have been adequately addressed? We are more than willing to continue the discussion if there are any lingering issues. Thank you sincerely for your efforts!

---

> > ### Comment · Reviewer_YGBs · 2023-11-22
> >
> > Thanks authors for explanations although I am not convinced. I prefer to keep my score.

---

### Official Review · Reviewer_WjUY · 2023-10-31

**Soundness:** 3 good
**Presentation:** 4 excellent
**Contribution:** 2 fair
**Rating:** 5
**Confidence:** 4

**Summary:**

This paper proposes a generic framework SUNG for sample-efficient offline-to-online RL, which introduces an optimistic exploration strategy via bi-level action selection to select informative actions for efficient exploration and develops an adaptive exploitation method with OOD sample identification to smoothly bridge offline RL and online RL objectives.

**Strengths:**

1. This paper is well-written and easy to follow.

2. The problem studied in this paper is important and has attracted increasing attention.

3. The experiment is thorough, and the authors compared SUNG against a large pool of recent methods.

**Weaknesses:**

This paper incrementally adds many existing techniques, making evaluating its contribution difficult. For example, the utilization of VAE for uncertainty quantification cannot distinguish SUNG from MANY offline-to-online or offline RL methods [1,2]. The bi-level action selection is a relatively heuristic strategy; the authors did not provide any theoretical analysis/insight into why it is effective, especially for the claim "we establish the ranking criteria for the finalist action set as uncertainty for value regularization-based methods and as Q value for other offline RL methods". Why does SUNG use Q value for other offline RL methods?

Moreover, SUNG introduces many new hyperparameters to be tuned. However, hyperparameter tuning has proven to be a challenging task in offline or offline-to-online RL.

Overall, SUNG is not an elegant and "simple" method.

[1] Zhou, W., Bajracharya, S., & Held, D. (2021, October). Plas: Latent action space for offline reinforcement learning. In Conference on Robot Learning (pp. 1719-1735). PMLR.

[2] Rezaeifar, Shideh, et al. "Offline reinforcement learning as anti-exploration." Proceedings of the AAAI Conference on Artificial Intelligence. Vol. 36. No. 7. 2022.

**Questions:**

The claim in section 4.3 " Note that offline RL methods do not suffer from state distribution shift during training, since policy evaluation only queries Q functions with states present in the offline dataset." seems to be incorrect in some model-based offline RL methods, e.g., MOPO, in which policy evaluation will query Q function with states generated by the model.

---post-rebuttal comment---
Thanks so much for providing the detailed responses. Unfortunately, I am not satisfied with them, especially those to Q2-4. I will reconsider the score if the authors can respond to my questions directly.

---

> ### Author Response · Authors · 2023-11-11
>
> **(Part 1/2)**
>
> ---
>
> We express our heartfelt appreciation to Reviewer WjUY for their detailed and valuable review. In response to the weaknesses and questions, we provide the following clarifications and improvements.
>
> > **Q1: Incremental contribution.**
>
> We want to highlight our contributions from the following aspects.
>
> **Firstly**, we mark that we identify the key question in offline-to-online RL as simultaneously addressing both challenges of constrained exploratory behavior and state-action distribution shift while eliminating the inherent conflict between exploration and exploitation. While prior research has attempted to address each challenge independently, tackling both challenges simultaneously is not a trivial task. For instance, some previous works (e.g., O3F) proposes efficient exploration strategy to mitigate the constrained exploratory behavior. However, we observe their divergence and sub-optimal performance when finetuning CQL agents, precisely due to this inherent conflict. Please refer to Appendix B for detailed analyses. In contrast, SUNG naturally unifies the solutions to both challenges with the tool of uncertainty, achieving the desired trade-off between exploration and exploitation. As a result, when combined with either TD3+BC or CQL, SUNG significantly surpasses previous SOTAs.
>
> **Secondly**, we point out that we develop novel techniques, such as bi-level action selection mechanism and adaptive exploitation with OOD sample identification, enabling SUNG to present a simple-yet-effective manner. Different from previous optimistic exploration techniques, we propose a bi-level action selection mechanism to mitigate the challenges of computation complexity and intensive task-specific hyper-parameter tuning, while achieving efficient exploration. In contrast to previous adaptive exploitation methods, SUNG builds on the insight that offline RL objectives can be typically interpreted as an online RL objective and a regularization term. As such, we develop a novel adaptive exploitation method with OOD sample identification to achieve the desired trade-off between performance and stability. Significantly, SUNG demonstrates robust superiority over previous SOTAs across a diverse set of hyper-parameters, as shown in Figure 3 and 4, which further enhances its practical applicability.
>
> **Thirdly**, we emphasize that we perform systematic empirical evaluations for SUNG. We combine SUNG with 4 backbone offline RL algorithms across 15 settings spanning MuJoCo and AntMaze domains, and we utilize 13 baselines for comparison. Experimental results demonstrate that SUNG significantly outperforms previous SOTAs. Furthermore, we conduct extensive ablation studies on each component to thoroughly verify the effectiveness of SUNG. Additionally, we also showcase SUNG's seamless integration with other techniques, such as ensemble Q learning and high update-to-data ratios, leading to superior finetuning performance.
>
> **Finally**, we also contribute a series of fresh insights to the offline-to-online setting, e.g., (1) Exploration with high Q-value actions leads to divergence when finetuning CQL, as elaborated in Appendix B. SUNG can stably finetune CQL agents by balancing the trade-off between uncertainty and Q value. (2) Fully online RL objectives fail in challenging sparse-reward tasks, as detailed in Appendix D.1. SUNG can achieve flexible trade-off between offline and online RL objectives, which enables stiching sub-optimal sub-trajectories and overcoming the sparse reward issue.
>
> **Overall**, in this paper, we aim to develop a simple, generic, sample-efficient offline-to-online RL framework as a good starting point for deriving strong empirical performance.
>
> > **Q2: Theoretical analysis/insight on bi-level action selection.**
>
> We want to highlight that the bi-level action selection mechanism is a practical solution for Eq. (7). It overcomes two challenges of solving Eq.(7): the difficulty in finding an optimal solution and the necessity for task-specific hyper-parameter tuning. Although we do not include theoretical insights of it, we empirically demonstrate the effectiveness of the proposed optimistic exploration strategy.

---

> ### Author Response · Authors · 2023-11-11
>
> **(Part 2/2)**
>
> ---
>
> >**Q3: Claim of ranking criteria.**
>
> The choice is underpinned by the following considerations:
>
> Firstly, we want to highlight the empirical finding that value regularization based offline RL methods (e.g., CQL) are only compatible with preference for high uncertainty, whereas other offline RL methods are compatible with preference for both high uncertainty and high value. We give a detailed analysis for the reason of this finding in Appendix B.
>
> Secondly, choosing the ranking criteria as "value first" implies that a low value of k indicates a preference for high value, whereas a high value of k indicates a preference for high uncertainty. Conversely, if the ranking criteria are set as "uncertainty first", the scenario is precisely the opposite. As such, by altering ranking criteria and k, the algorithm allows for the customization of preferences between uncertainty and Q value to varying extents.
>
> Thus, irrespective of the chosen ranking criteria, we possess the flexibility to adeptly adjust k to attain the desired preference. Consequently, we select the aforementioned criteria to fix the tuning range within a small value of k, only aiming for consistency.
>
> >**Q4: Concern on hyper-parameter tuning.**
>
> We want to clarify that SUNG does not require extensive hyper-parameter tuning; instead, it only demands minimal effort in that regard. As demonstrated in Figure 3, SUNG consistently outperforms the best baseline for any choice of k, with the optimal D4RL score achieved at k=10/20. Similarly, as illustrated in Figure 4, SUNG consistently outperforms the best baseline when p is appropriately set within the range of [5, 10], with the highest D4RL score obtained at p=5. Thus, SUNG achieves robust performance without the need for intensive hyper-parameter fine-tuning, which aligns with our initial motivation to develop a simple-yet-effective offline-to-online RL algorithm. We only tune these two hyper-parameters of SUNG in this paper, i.e., finalist action set size k and OOD sample percent p.
>
> >**Q5: Claim for state distribution shift.**
>
> Thanks for pointing out this ambiguity. You are right. Here, offline RL methods refer to model-free ones. We make this statement clearer in the revised version.
>
> ---
> We hope that our response could address your concerns and that you would consider increasing your score.

---

> ### Author Response · Authors · 2023-11-21
>
> Dear reviewer WjUY, as the deadline for author-reviewer discussions is approaching, could you kindly confirm if your concerns have been adequately addressed? We are more than willing to continue the discussion if there are any lingering issues. Thank you sincerely for your efforts!

---

### Official Review · Reviewer_K3Su · 2023-10-31

**Soundness:** 3 good
**Presentation:** 3 good
**Contribution:** 2 fair
**Rating:** 5
**Confidence:** 4

**Summary:**

The paper proposes a unified methodology for offline to online reinforcement learning based on an out of distribution sample identifying mechanism that employs a variational autoencoder for density estimation and ultimately to quantify uncertainty. The authors propose the SUNG (simple unified uncertainty guided) framework to work on top of most model-free offline RL algorithms & evaluate the approach with CQL & TD3+BC as base algorithms. Since a key limitation when moving from offline to online learning is the conservatism that hinders collection of new data, the algorithm employs the concept of optimism in the face of uncertainty so that the policy should in the online phase choose actions that have both high value and high uncertainty. Since the distribution shift induced by this exploration scheme makes value estimation harder and can hinder learning, an adaptive uncertainty guided exploitation scheme is introduced, which regularizes the policy based on the samples' OOD-ness.
The empirical evaluation compares SUNG with two base algorithms against multiple other baselines on D4RL tasks and finds SUNG to outperform prior works substantially.

**Strengths:**

Offline to Online learning is still a relatively new discipline and the authors appear to have found a simple yet effective method to outperform prior works. Selecting actions optimistically in the face of uncertainty seems like a good exploration strategy for O2O, since it's been proven to work in prior works on other exploration tasks. Especially the fact that the method is compatible with many offline RL algorithms that can be used under the hood as base algorithm appears to be a practical advantage.

**Weaknesses:**

I find the formulation of the SUNG framework a bit counterintuitive: The authors mention that they want to have high-uncertainty actions, yet at the same time they only sample "near-on-policy actions for exploration", which appears contradicting. Further, during the optimization / policy improvement part (green arrows in fig 1), the same percentage p of the batch is always labeled as OOD, which is not consistent, since the absolute uncertainty value at which a sample could be labeled OOD can vary (it could even be the case that the same sample is sometimes OOD and sometimes not, depending on the other samples in the batch).

Generally, I have a bit of an issue with the additional OOD detection in the adaptive exploitation - the regularizer in offline RL algorithms is mostly already doing OOD detection (often also based on some measure of uncertainty), so it should already automatically detect low and high uncertainty samples and thus assign low or high regularization accordingly. While this is normally not done in such a binary nature, I don't see how the adaptive exploitation scheme is not doing pretty much the same thing again. It seems to just amplify the regularization tendency that is already there. The emperical ablation in fig 2d shows it to be important, but to me it is unclear why that is the case - is the normal regularization simply not yet enough? or is the VAE based OOD detection in some qualitative way better than the base algorithms own regularization?

In the exploration scheme, where actions are ranked & selected based on value and then sampled based on uncertainty (or the other way around), you mention the possibility to adapt the trade-off preference between value and uncertainty - what do you mean by that? Would you adaptively change the trade-off during the training process or is it something you set in advance (and based on what information)? I'm also not sure whether you mention which way around you ended up doing filtering + sampling (first value or first uncertainty or maybe some combination) and why.

In the empirical evaluation I'm assuming you report final performances (i.e. after the 100k environment steps). Prior works that you mention (like Cal-QL, AWAC) have shown that during O2O, algorithms commonly suffer from immediate performance drops right after online learning starts due to distribution drift - it would be extremely interesting to see how SUNG performs in this context, which is why another performance metric reporting during the beginning of online learning could be useful or directly plotting the test returns over training time.

I believe some other prior works that are currently not contained should also be considered in the related work section:

[1] Swazinna, P., Udluft, S., & Runkler, T. (2021). Overcoming model bias for robust offline deep reinforcement learning. Engineering Applications of Artificial Intelligence (EAAI), 104.

[2] Ghosh, D., Ajay, A., Agrawal, P., & Levine, S. (2022). Offline rl policies should be trained to be adaptive. ICML 2022

[3] Hong, J., Kumar, A., & Levine, S. (2022). Confidence-Conditioned Value Functions for Offline Reinforcement Learning. ICLR 2023

[4] Swazinna, P., Udluft, S., & Runkler, T. (2022). User-Interactive Offline Reinforcement Learning. ICLR 2023

[1] introduces MOOSE, which is a model-based offline RL method that uses a VAE as a regularizer. While your adaptive exploitation scheme is using it in a different way, I still believe it should be mentioned. The works [2-4] are also concerned with offline to online learning, just that their online phase is a little shorter and their adaptations thus look a little different than the one you consider. Still, when thinking about O2O they are closely related and should be considered.

More a sidenote: Fig. 2b is labeled "Optimistic Exploration without Uncertainty". As far as I understand, this method is greedily selecting best actions based on value - it is unclear to me how that is optimistic (i.e. optimisim in the face of what) so maybe the different ablations should be named more clearly.

I realize the weakness section is a bit lengthy, but I think your paper has merits and I am prepared to increase the score if you are able to address my concerns.

**Questions:**

See weakness section

---

> ### Author Response · Authors · 2023-11-11
>
> We would like to sincerely thank Reviewer K3Su for providing the constructive review, and we respond to the weaknesses and questions as below.
>
> > **Q1: Contradiction between high-uncertainty and near-on-policy actions.**
>
> As shown in Eq. (7), SUNG involves selecting actions with both high Q values and high uncertainty to guide exploration. Specifically, we restrict our exploration to near-on-policy actions to prevent significant deviations. This restriction offers two key advantages:
>
> (1) Ensuring finetuning stability: By exploring near-on-policy actions during finetuning, we maintain stability in the learning process. The policy improvement relies on experiences collected during exploration. Deviating too far from the current policy could lead to abrupt and undesirable changes, rendering the learning process unstable and less sample-efficient.
>
> (2) Avoiding inaccurate Q value and uncertainty estimation: Sampling actions far from the on-policy actions may result in less reliable estimates of Q values and uncertainty. This unreliability has the potential to throw the exploration direction out of control, hindering the overall effectiveness and efficiency of exploration.
>
> > **Q2: The absolute uncertainty value at which a sample could be labeled as OOD can vary.**
>
> We fully agree with your perspectives. SUNG detects OOD samples via percentage-based filtering strategy out of simplicity. This approach is intentionally designed to minimize the complexity associated with hyper-parameter tuning. However, we want to highlight that empirical results demonstrate the effectiveness of SUNG, as evidenced by the SOTA finetuning performance achieved by SUNG through the proposed adaptive exploitation. We acknowledge that there is room for future exploration in refining adaptive and precise identification of OOD samples.
>
> > **Q3: Why introducing additional OOD detection in the adaptive exploitation.**
>
> Although many offline RL methods integrate OOD detection in the adaptive exploitation, they still necessitate heavy conservatism in the objective to avoid the extrapolation error. In this paper, SUNG introduces OOD sample identification to adaptively relax such conservatism for higher sample efficiency during online finetuning. It is noteworthy that certain offline RL algorithms, including TD3+BC, CQL, and SPOT, lack explicit OOD detection. Hence, the proposed adaptive exploitation holds significant importance for these methods to attain desirable fine-tuning performance.
>
> > **Q4: Explanation on trade-off between value and uncertainty.**
>
> In the suggested optimistic exploration strategy, choosing the ranking criteria as "value first" implies that a low value of k indicates a preference for high value, whereas a high value of k indicates a preference for high uncertainty. Conversely, if the ranking criteria are set as "uncertainty first", the scenario is precisely the opposite. As such, by altering ranking criteria and k, the algorithm allows for the customization of preferences between uncertainty and Q value to varying extents. This flexibility empowers users to finely adjust the trade-off according to the specific requirements of their realistic scenarios.
>
> > **Q5: How to select the ranking criteria?**
>
> We detail this problem in Appendix A.2 and A.3. Specifically, we choose uncertainty first for value regularization based offline RL methods, whereas Q first for other offline RL methods. This choice is underpinned by the following considerations. Firstly, we want to highlight the empirical finding that value regularization based offline RL methods (e.g., CQL) are only compatible with preference for high uncertainty, whereas other offline RL methods are compatible with preference for both high uncertainty and high value. We give a detailed analysis for the reason of this finding in Appendix B. As articulated in Q4, irrespective of the chosen ranking criteria, we possess the flexibility to adeptly adjust k to attain the desired preference. Consequently, we select the aforementioned criteria to fix the tuning range within a small value of k, only aiming for consistency.
>
> > **Q6: Other evaluation metrics.**
>
> Since this paper focuses on the improving the sample efficiency of offline-to-online RL, we choose the final average performance of 100k environment steps as the evaluation metric in the main body of this paper for clear comparison. Note that we also provide the full training curve for 30 settings in Appendix D.4 to further demonstrate the effectiveness of SUNG.
>
> > **Q7: Some related works.**
>
> Thanks for providing these related works! They are all included for discussion in the revised version.
>
> > **Q8: Label of "Optimistic Exploration without Uncertainty"**
>
> Here, the optimism refers to exploration with optimism in the face of Q value. We give detailed experiment settings of the ablation study in Appendix D.5.
>
> ---
> We hope that our response could address your concerns and that you would consider increasing your score.

---

> > ### Comment · Reviewer_K3Su · 2023-11-21
> > **Rebuttal Response**
> >
> > Dear authors,
> >
> > thank you for your detailed answer to my questions and concerns. I have only a single follow-up question:
> >
> > You make a distinction between conservatism and OOD detection (see your response to question 3). I would argue the two are very closely related, i.e. conservatism is usually enforced by trying to regularise the policy to only perform actions that can be considered in distribution and filtering out everything else - this seems to be pretty much the same as detecting out of distribution actions and avoiding them. Can you explain to me where you see the key difference between the two, and thus one of the key added advantages your method brings to the table?

---

> ### Author Response · Authors · 2023-11-21
>
> Dear reviewer K3Su, as the deadline for author-reviewer discussions is approaching, could you kindly confirm if your concerns have been adequately addressed? We are more than willing to continue the discussion if there are any lingering issues. Thank you sincerely for your efforts!

---

### Meta-Review · Area_Chair_myhs · 2023-12-13

**Metareview:**

In this paper, the authors proposed an algorithm for offline-to-online reinforcement learning, with vanilla offline Q and policy learning and VAE for uncertainty training in offline phase, and an exploration and OOD adaptive exploitation component in online phase. The method is empirical verified.

**Justification For Why Not Higher Score:**

There are several concerns raised by the reviewers:

1, The formulation of the SUNG framework is not well-justified. The proposed exploration with VAE and OOD exploitation are both heuristic. Several algorithm steps, e.g., the top-k action selection, do not have formal justification.


2, In the empirical comparison, there is no hyperparameter tuning reported, and it will be also interesting to report the performance curve during offline to online training.

**Justification For Why Not Lower Score:**

N/A

---

### Decision · Program_Chairs · 2024-01-16

Reject